# COVID-19 impacts equine welfare: Policy implications for laminitis and obesity

**Ashley B. Ward[1,2], Kate Stephen[1], Caroline McGregor Argo[1], Patricia A. Harris[3], Christine A. Watson[1], Madalina Neacsu[2], Wendy Russell[2], Dai H. Grove-White[4], Philippa K. Morrison[1] ***

**1** Scotland's Rural College, Aberdeen, United Kingdom, **2** The Rowett Institute, University of Aberdeen, Foresterhill, Aberdeen, United Kingdom, **3** Equine Studies Group, WALTHAM Petcare Science Institute, Leicestershire, United Kingdom, **4** Faculty of Health and Life Sciences, University of Liverpool, Wirral, United Kingdom

* Philippa.Morrison@sruc.ac.uk

## Abstract

The COVID-19 pandemic continues to impact human health and welfare on a global level. In March 2020, stringent national restrictions were enforced in the UK to protect public health and slow the spread of the SARS-CoV-2 virus. Restrictions were likely to have resulted in collateral consequences for the health and welfare of horses and ponies, especially those at risk of obesity and laminitis and this issue warranted more detailed exploration. The current study utilised qualitative methodology to investigate the implications of COVID-19 related policies upon equine management and welfare with a focus on horses and ponies at risk of laminitis and obesity. Twenty-four interviews with five sub-groups of key equestrian welfare stakeholders in the UK were conducted between May and July of 2020 to understand the challenges facing equine welfare in the context of laminitis and obesity susceptible animals. Thematic analysis revealed lockdown-associated factors with the potential to compromise welfare of horses and ponies at risk of obesity and laminitis. These included: disparate information and guidance, difficulties enacting public health measures in yard environments, and horses having reduced exercise during the pandemic. Positive examples of clear and consistent information sharing by farriers were reported to have improved horse owner understanding of routine hoof care during lockdown. Analysis suggested that the recommendations for supporting the management-based needs of horses under reduced supervision were not clearly defined, or were not sufficiently disseminated, across the equine industry. These findings support the development of guidelines specific to the care of horses and ponies at risk of obesity and laminitis through collaborative input from veterinary and welfare experts, to reduce the negative impacts of future lockdown events in the UK.

## Introduction

On January 1st, 2020, the World Health Organization (WHO) activated an emergency response framework to address reports of an atypical respiratory disease concentrated in

**Data Availability Statement:** The data collected for the present study may contain details which could potentially lead to the identification of study participants. As such, requests for access to this data may be sent to SRUC's Data Protection

Officer. Address: Scotland's Rural College, Executive Office, Peter Wilson Building, Kings Buildings, West Mains Road, Edinburgh EH9 3JG. Telephone: 0131 535 4432 E-mail: dpo@sruc.ac. uk.

**Funding:** This study was funded by Mars Petcare and is part of a PhD studentship funded by the Scottish Funding Council Research Excellence Grant (REG). Authors WR and MN receive salary support from the Rural and Environment Science and Analytical Services Division (RESAS). With the exception of PH (employed by the funding organization), the funding organization did not have any additional role in the conceptualization, methodology, investigation, data curation, formal analysis, decision to publish, or preparation of the manuscript. PH was involved in study design, data interpretation, and manuscript preparation.

**Competing interests:** Co-author PH is employed by the funding organization. This does not alter our adherence to PLOS ONE policies on sharing data and materials. All other authors declare that they have no competing interests.

Wuhan, China [1]. As case rates of this highly infectious SARS-CoV-2 virus (termed COVID-19) accelerated, the first pandemic since that of the 2009 influenza A (H1N1) virus was declared on March 11[th], 2020 [2]. In response to rapidly rising rates of infection in the United Kingdom (UK), the country was placed into a national "lockdown" on March 23rd, 2020 [3].

Lockdown involved the implementation of public health measures that restricted social interaction, non-essential work activities and travel, which inevitably resulted in significant disruption to the daily routines of the general public. Travel was permitted in circumstances that involved the necessary provision of care to animals, which included caring for companion equines. It is estimated that between 32% [4] and 59% [5] of horses are housed at establishments, termed livery yards, requiring owners to travel in order to provide for their horse's needs. To align with safe working practices during COVID-19, these equestrian businesses were required to adopt measures to protect public health, enable social distancing and promote hygiene.

Decisions regarding public health measures to be employed fell to the establishment proprietors. During lockdown and the associated "infodemic" [6], many sectors were bombarded with COVID-19 related information from multiple sources and platforms with varying degrees of credibility. Some horse owners may have looked for guidance from sporting, welfare or veterinary authorities that preside over different aspects of equine leisure and management in the UK; the sheer number of which makes it difficult to trace decision making to a specific source. This phenomenon appears to be an affliction of the equestrian industry across the world. Challenges to information dissemination in the industry in Australia were suggested by Schemann et al. (2012) to have been exacerbated by the multitude of pursuits and disciplines of equestrian sport [7]. Considerable efforts to address disparate messaging to industry stakeholders, specifically to minimize confusion around the pandemic, were promptly undertaken by Greene et al. (2020) in the US [8]. However, in the UK, it is not yet understood how decisions regarding equine health and management during the pandemic were informed across the industry, particularly in the case of establishments housing multiple animals.

Recent research has shown that the reduction in routine veterinary care during lockdown was a source of concern for owners of horses and ponies [9] but this worry may have been exacerbated for owners of animals that were at increased risk of management associated clinical conditions. Such is the case in horses and ponies native to the British Isles, which are understood to be at increased risk of obesity and endocrinopathic pasture associated laminitis-a common and potentially lethal pathology of the hoof which can be triggered by diet-induced perturbations of carbohydrate metabolism [10]. Managing the risk of laminitis requires the provision of appropriate nutrition, exercise and hoof care [11] through involved collaboration between the horse owner, veterinarian and farrier [12]. The goal of managing susceptible horses is to prevent exposure to suspected laminitis risk factors and triggers; obesity, pasture rich in non-structural carbohydrates (NSC), and insulin dysregulation [13, 14], and to preserve the optimal hoof conformation. National COVID-19 related restrictions are likely to have disrupted implementation of preventative management practices in place for those horses at risk of obesity and laminitis. Indeed, in a recent publication, van Dobbenburgh and De Briyne (2020) highlighted significant concerns raised by veterinary authorities over the welfare of companion animals, including horses, given the reduced access to veterinary care, the financial burden of economic stressors, and the restrictions to movement faced by all [15]. Given the overwhelming impact that COVID-19 has had upon the economy and public health in the UK, it is important to widely explore the social mechanisms through which equestrian industry stakeholders were impacted by the pandemic, in order to assess the repercussions on equine welfare.

Qualitative interpretation of the social impact of a global event, such as the COVID-19 pandemic, can offer insight into the socioeconomic and psychological consequences of the event on a given group of individuals [16]. Although government guidance was developed and distributed to specific governing bodies in the equestrian industry, such as the British Horse Society (BHS) and Horse Scotland, the direct impact of the resulting advice has not yet been studied. Documenting the experience of horse owners and industry professionals could provide insight into personal and professional experiences of the restrictions imposed during the nationwide lockdown, and thus enhance our understanding of how equine welfare may have been challenged as a result. Interrogation of the pandemic's impact within the social context of equine and laminitis management may reveal novel phenomena experienced by stakeholders, which could serve to inform future policy development and decision making regarding the care of equids and other externally housed animal species during national and global emergencies.

The current study was designed to investigate the implications of COVID-19 related policies upon equine management and welfare with a focus on laminitis and obesity. Key objectives of the study were to assess the impact of the pandemic on the management of laminitis susceptible horses and ponies, to identify challenges faced in implementing COVID-19 based guidance, and to identify areas of decision making and policy development which could undergo improvement in future pandemic or emergency scenarios. To gain a multi-faceted insight, five groups of key equine stakeholders were interviewed using a semi-structured approach in line with grounded theory methodology. It is anticipated that the results from this study will provide a key reference during conversations regarding public health measures that may impact equine welfare in the UK.

## Materials and methods

The study was designed and conducted within the methodological framework of hermeneutic phenomenology. The grounded theory aspect of the methodological approach was consistent in the inductive, comparative approach to analysis, coding of text and verification of themes which arose from the data [17]. Interviews were designed with pre-defined questions to guide discussions. However, in line with research methods utilising an iterative approach, interviewees were able to lead conversation in areas relating to the pandemic, laminitis, and obesity meaning that as the research evolved, certain concepts were not discussed across all interviews. The primary researcher, a PhD student, received training in qualitative research methods prior to and throughout the study design before conducting a pilot interview which was recorded and assessed to determine the appropriateness of questions, interview style and overall approach. A researcher with expertise in social sciences and conducting qualitative research (KS) was consulted throughout the design of the study and provided structured feedback for interview improvement after evaluating the pilot interview.

Ethical approval from the SRUC social sciences ethics review committee was obtained on 01/05/2020. Recruitment was conducted by word of mouth, social media, organisational connections, and employing the assistance of a "gatekeeper" to access the target community. Participants consented to take part in the research through completing a tick-box online participant information sheet, documenting the purpose and aims of the study, as well as the questions that would be posed. This online document also included a statement of data protection conforming with the GDPR guidelines of SRUC and served as a record of signed consent from the participant. Participants were also asked for their verbal consent to take part in the study prior to the initiation of interviews. This was captured within the recorded audio for each interview.

Semi-structured interviews were conducted through recorded telephone calls with participants based in the Aberdeenshire region (n = 22), except for two interviewees in England (managers of equine welfare centres in Somerset and Blackpool). The sampling method employed at outset was a purposeful, pragmatic approach involving convenience, but heterogenous, sampling. Such a strategy ensured that a variety of perspectives from key sub-population informants were included, and that theoretical saturation could be attained. Further participants were recruited as opportunity arose through connections with interviewees. A minimum of five participants from each group was sought from the outset of sampling. This figure was selected as a minimum number estimated to be necessary to glean sufficient information from within each group studied, based upon estimations of the probability of observing multiple codes within each group [18]. Sampling ceased after three months had elapsed from the time of the most stringent lockdown guidelines being lifted to minimize the impact of the passage of time upon the interviewees' memory and perception of the impact that the lockdown event had on their equine management or practice. The five selected groups were; native-breed horse or pony owners with animals kept at home (n = 6), native breed horse or pony owners with animals kept at livery (n = 5), equine veterinarians (n = 5), registered farriers (n = 4), and equine welfare centre mangers (n = 4). The inclusion and exclusion criteria used to determine suitability of participants for the study can be found in S1 Table. Livery yard stakeholders were based at different livery yards within the Aberdeenshire region. These premises were varied in terms of facilities available and type of livery packages offered, and the horse owners included leisure riders, as well as those participating in competition to county level. More detailed interviewee characteristics are available in S2 Table.

In total, 13 hours of interviews were conducted across a period of two months, from May-July 2020, with a mean interview length of 32 minutes (min = 17 mins, max = 45 mins). Detailed, handwritten notes were compiled during each interview, adding contextualization to the audio recording, such as intonation, further context, hesitation and enthusiasm. Audio recordings of the telephone interviews were made using Skype for Business for Office 365, and audio files stored in the SRUC One Drive in a password protected file. Audio files were transcribed verbatim by the primary researcher, and descriptive coding was conducted throughout the data collection period. Additional areas of interest which emerged during interviews were added to the list of prompts used by the interviewer. This iterative process involved probing for information to increase sampling efficiency, with the aim of maximising thematic saturation [19]. The set of predefined interview questions and the overarching understanding sought through the questions are outlined in Table 1.

Data generated through interviews were uploaded to SRUC OneDrive folders as MP4 files, transcribed *verbatim* using VLC Media Player (3.0.8 Vetinari) for playback, and Microsoft Word. Transcripts stored within word files were then uploaded to NVivo Windows (1.0). At the point of transcription, all identifiable details in the interviews were removed from the files. To de-identify transcripts, labels based on the participants' grouping were assigned to each interview file and were used to refer to interviewees thereafter. The full transcripts can be found in S1 File.

Data were organised and interpreted using an iterative coding process directed by a hermeneutic approach to analysis [20]. Descriptive codes were assigned to units of text which appeared meaningful. Meaningfulness was determined by the connection of the text unit to the subject of the management of obesity and laminitis, or by the demonstration of a viewpoint, opinion, example or stance of the interviewee on any subject. Categories of common or significant codes were then developed based upon either the frequency of expression, or the weight of emotion or importance of the concept to the interviewee [21]. An example of this process can be viewed in S3 Table. This process was repeated and updated several times across

**Table 1. Predefined interview questions with sub-topic probes which interviewees were asked.**

| Broad topic question | Sub-topic |
|---|---|
| **What preventive measurements have become difficult to enact?** | • Routine<br>• Feeding<br>• Exercise<br>• Pasture management |
| **What are the primary challenges to managing horse health faced by horse owners/vets/professionals during the pandemic?** | • Social distancing<br>• Pasture management<br>• Time<br>• Finances |
| **What are your greatest concerns for horse welfare?** | • Laminitis<br>• Day to day management<br>• Waste management<br>• Recognition of injury / distress<br>• Obesity<br>• Restrictions on movement<br>• Feed availability—too much too little (awareness)<br>• Farrier<br>• Emergency treatment<br>• Foaling |
| **Do you have plans in place to minimize/address your concerns?** | • Illness / emergency<br>• Laminitis management |

the data set as this expanded. Annotations were recorded to incorporate notes taken during the interviews, as well as to describe meaningful relationships between information across the data. Descriptive codes were then grouped into subject area in tabular form, before being coded using more meaningful, inductive codes developed from the grounded theory approach, allowing the content of the text to instruct the following phase of coding. Themes were identified and named, before being re-assessed and re-named following scrutiny of their appropriateness to represent the phenomenon underlying the text. Analysis was carried out by the primary researcher, and tables of units, codes, sub themes and themes were recorded and shared amongst the research team to undergo discussion of appropriateness at multiple stages during the data analysis phase of the study. Such reviews were conducted to improve the validity of the theory being drawn from the interview transcripts, whilst maximising efficiency amongst the group.

## Results

Content and thematic analysis revealed four consistent themes associated with the implications of governmental and industry issued policy upon equine welfare and management. These were: 1) Challenges around guideline interpretation, (2) Implication of public health measures on routine preventative care for laminitis and obesity, (3) Outcomes of minimising risk of physical injury, and (4) Negative impact of the pandemic upon mental health. Themes and sub-themes that were extracted from interview transcript data are presented in S1 Fig. The authors considered that detailed exploration of the theme of mental health during the pandemic was out-with the scope of the present article and the data will be presented in a separate manuscript.

The following sections present the themes and sub-themes in more detail. Firstly in relation to the way in which the interpretation of guidelines affected stakeholders within the equestrian community, followed by the way in which these guidelines impacted on the management of horses and ponies at greater risk of obesity and laminitis, and finally the experiences of interviewees regarding minimizing the risk of physical injury.

## Challenges around guideline interpretation

The interpretation of the guidelines disseminated from Government and industry level was a prominent area of concern across many of the interviews. The impact that information had on the welfare of horses and ponies at risk of obesity and laminitis was dependent upon the interpretation of those guidelines, and as such, led to extensive discussion around which practices were appropriate and which were not.

**Multiple sources of information.** For horse owners, a significant obstacle to decision making arose from difficulties in identifying a single source of information for COVID-19 guidance that was relevant to their specific situation. Horse owner interviewees identified several authorities that could be consulted regarding decision making during the pandemic, including: livery yard owners, the British Horse Society, local veterinary practices, Horse Scotland, British Equestrian Federation and British Eventing. Welfare centre managers referenced a further two authorities: the National Equine Welfare Council, and World Horse Welfare. There was no consensus over which was preferable, or which was the definitive authority designated to provide information to horse owners and managers. The following quote from a farrier aptly summarises the overall perceptions of information sources for horse owners:

> "I think if bigger governing bodies could have given information to horse owners, that would have been helpful. There was a lack of cohesive information. I think if there had been a clear authority that said, "relax, here's what you are doing", then that would have been better. People seemed to take information from lots of different sources–like the discipline specific bodies were giving information to some, and no overall body made collaboration difficult."
>
> –F3.

In contrast, farriers found their guidance concise and consistent. All interviewees in this group identified governmental messaging, the Farriers Registration Council (FRC) and the British Farriers and Blacksmiths Association (BFBA) as the three sole sources of information regarding their working practices during the pandemic. Interestingly, the EWC group perceived the multiple sources of information as less of an issue, but as having the potential for positive impacts on information exchange. One welfare centre manager outlined the connections between the organisations and highlighted the veterinary associations as the welfare centre's ultimate point of reference for information, saying:

> "you have NEWC, the National Equine Welfare Council, of which we are part of. That is basically equine charities which have all come together. . .. although we all work individually, we all know what each other are doing at one time. Obviously, the British Equestrian Federation, and BEVA, the veterinary advisory groups as well. . .our head office keeps abreast of exactly what they are going to issue, and we have to follow their guidelines."
>
> –WCM1.

Veterinarians consulted governmental advice, and that of the British Veterinary Association (BVA) and the British Equine Veterinary Association (BEVA) for their information.

Although the number of sources wasn't considered as overwhelming for this group, the slow release and lack of practical detail was highlighted as a challenge.

**Guidelines lacking detail.** The guidance regarding public health measures was considered superficial and was thought to lack sufficient detail to address biosecurity during aspects of veterinary procedures and general horse management. Ambiguous areas not detailed in guidelines included how to practice social distancing during dangerous scenarios whilst handling horses, and the exact requirement for PPE upon yard environments. Details regarding the transmission of the SARS- CoV-2 virus through contact with equipment was seen as sufficient, however the cost upon finances and staff time of sustaining disinfection protocols for extended periods of time were, "glazed over" (quote from HL2), and did not evolve as time passed.

Interviewees noted the unexpectedly prolonged experience of lockdown, with reference to the fact that the majority of guidance targeted public health requirements during the strict lockdown period, but guidance did not develop as the country progressed out of lockdown. The veterinary industry was seemingly under significant pressure to keep up with the case load to avoid being overwhelmed once restrictions were lifted. Equine veterinarian V4 discussed the difficulties associated with putting off routine work:

> "my feeling was that if we wait until June to do all of this routine work, we are going to have so much of a backlog that we will physically not be able to keep up with that, and stay heathy, or even take the extra two minutes to put a mask and gloves on"
>
> –V4.

The lack of pre-emptive guidance as the pandemic progressed caused frustration across the sample. Guidance from equestrian bodies was said to be too slow in relation to the government guidance issued, which left scope for agencies to subjectively interpret government advice–leading to inconsistent information. Equestrian and veterinary governing bodies were frequently urged by horse owner groups and veterinarians to, "sing from the same hymn sheet" (quote from V4). Situations were described in which businesses lost clients to local competitors that offered services deemed to be out of line with government lockdown guidelines. Such events caused frustration, resentment and conflict within the equestrian and veterinary communities. Furthermore, the variation in biosecurity measures undertaken by local yards left veterinarians feeling that, "you never know what you're going into", (quote from V4) which heightened concern over the possibilities for the spread of the virus.

There were also points in conversation where veterinarians indicated that they felt prepared for the pandemic in a sense, due their training in infectious disease and epidemiology.

> "I have no doubt that the RCVS (Royal College of Veterinary Surgeons) and BVA (British Veterinary Association) have been setting guidance, and also my employer who is a corporate employer, they would give out instructions every day. . . I was just following the simple biosecurity measures that we are always aware of,"
>
> - V2.

**Positive impact of "good" guidance.** The three characteristics of positively received guidance were: a credible, authoritative source; a clear structure; and continuity. There were two instances of seemingly highly effective information dissemination strategies highlighted by industry stakeholders. The first was a single account of a livery yard which issued a comprehensive outline of the plan for the business:

"Early on they devised an 8-phase plan for the yard, so it was all set out. We knew what would be happening on a date by date basis. The goalposts moved a little bit, but they had thought everything through to when furloughed staff would be coming back, when we would be starting up lessons again."

–HL-6.

Having a clearly structured set of guidelines in place with the flexibility to adapt for individual circumstances appeared to result in confidence and a sense of relief in interviewees who discussed the positive guidance they had received. Where governing bodies and authorities had adopted a firm and clear stance on practices that stakeholders should adhere to, there was a positive and constructive response.

Message continuity in the industry was seen to be achieved by the local farriers. Here, a horse owner highlighted the positive impact of such consistency:

"it was very clear that (the Farrier Council) had written something for all of the farriers to post on their farrier pages. So, for our farriers, they all had the exact same wording. So, I am guessing that somebody from some governing body has said to them, here is an example for something you could put up, it would be great if we could all do the same. So that was good because it gave the same message across the board."

–HL2.

The guidance adhered to by farriers was delivered by the BFBA and appeared to provide farriers with sufficient confidence that their practice remained within public health advice.

"Social distancing- I employed based on the government guidance, and the Farriers Registration Council gave a really good traffic light system approach."

- F2.

The traffic light system utilised by farriers was a risk-based system published by the BFBA directed to provide registered farriers with guidance determining the urgency of farrier care required on a case-by-case basis. A pragmatic approach was often described by the farriers interviewed, who frequently referred to using, and expecting others to use, "common sense" (quote from F3), in order to preserve safety during farrier visits and with regard to the pandemic in general.

## Implications of public health measures on equestrian livery yards

The public health measures that interviewees most commonly discussed in the context of livery yards were social distancing, hand washing and hygiene, as well as the use of PPE. Another important topic of conversation in relation to the practices one could implement to follow national lockdown guidelines was turning horses away. To "turn away", is the practice of increasing a horse's time in a field environment, whilst decreasing structured exercise and feeding regimens. In efforts to align with governmental public heath recommendations in the livery yard context, the three practices implemented or experienced by interviewees were turning horses away, altering veterinary and hoof care regimens, and restricting access to livery yards.

**Turning horses away.** Descriptions of concerns for equine health during lockdown frequently involved the subject of animals having increased access to grazing due to being turned

**Fig 1. Perceived and actual reasons for turning horses away during the COVID-19 lockdown.** Horse owners, veterinarians, farriers and welfare centre managers discussed their perceived reasons that horses may have been turned away to grass during lockdown. The confirmed reasons include actual reasoning employed by interviewees, or their recollections of reasoning used by others, for turning horses away to grass despite the animals' susceptibility to laminitis and obesity.

away. The interviewees provided a range of perceived reasons for animals being turned away to grass: financial difficulty, staff availability and workload, and behavioural benefits.

The first of the perceived reasons that owners might elect to turn animals out to grass included financial difficulty associated with the pandemic. There was an expectation that horse owners who had experienced furlough or job loss may be forced to turn horses out into fields as a cheaper alternative to stabling. This phenomenon was not directly experienced or witnessed by any interviewee, although the majority expressed concern that financial hardship would be the cause of increased abandonment of horses, "post lockdown". Perceived and actual reasons that interviewees discussed regarding turning horses away are presented in Fig 1.

Most commonly, interviewees predicted that horses would be turned away due to owners being restricted from accessing their yard to provide care- resulting in an unmanageable work-load for a reduced number of retained yard staff. It should be noted that one interviewee feared the opposite effect: that horses would spend increased time stabled which would have detri-mental effects on their mental wellbeing. Interviewees gave direct examples of horses being turned away under the "yard closure" scenario. A direct experience was recounted of a livery yard / riding school which had furloughed staff members, leaving insufficient staff numbers to care for stabled horses. In this instance, horses which did not normally have access to unre-stricted grazing had been turned away onto pastures (newly acquired for this purpose), and laminitis had affected two horses:

> "We had two (horses) come in (to be stabled) with fairly serious laminitis in the past two weeks. . . Because we have had to furlough a lot of people and just keep one person working, a lot of them have had to be turned away onto grass- probably for the first time in a couple of years for some of them."

> –HL 6.

Welfare centre managers tended to support decisions to turn horses away in discussions surrounding the wider equine community but emphasized that a "risk assessment" should be

adopted when deciding whether this practice would suit individual animals, taking into account behavioural characteristics and disease risk.

Horse owners with their horses at home did not discuss having to make the decision to turn their own horses away, however an account was provided of an instance of a horse owner effectively abandoning a horse in a field due to being unable to leave the house to attend it. An interviewee described acquiring an obese native pony whose owner suffered from severe health concerns and whose family had experienced death as a result of the SARS-CoV-2 virus. In this instance, the owner was self-isolating and was unable to drive to the field where the horse was kept. This resulted in the animal being turned out for an extended period of time, subsequently becoming obese and being rehomed with the interviewee.

> "she [the owner] had acute asthma and had to self-isolate herself, and the horse had been in the field—lush green field with the sheep. . . so I said, I am happy to take your pony indefinitely and stick her in with mine. . .It was in a field mile away and she would have to drive to see it—she was worried about her horse . . .When I got [the horse] home, she was like an elephant"

> –HH1.

Equine veterinarians expressed frustration at the practice of increasing horses' access to grazing without measures to limit intake; particularly where animals were at risk of obesity or laminitis. Three of five shared surprise at the extent of weight gain they had witnessed, expressed clearly by one vet who stated:

> "I think one big problem we had with respect to laminitis was that because the weather was good, quite a few yards- their solution to not having liveries up was just to chuck all the horses out in the field. And some horses I have seen post lock down, I have never seen them so fat, ever."

> –V4.

Across interviews, equine obesity was identified as the most worrying threat to equine welfare in the region. However, interviewees often made a point to note that equine obesity was an ongoing issue in the industry regardless of the pandemic. Interestingly, although horse owners identified obesity as an alarming trend occurring in the equestrian community, the majority of horse owners also referred to their own animals as being overweight, although terms such as "tubby" and "chunky" were employed when referencing their own animals, whereas, "fat" and "obese", were more likely to be used to describe animals that individuals did not own. This was not exclusively the case, however, and owners tended to be self-punishing where they felt they had "allowed" their horses to become overweight in the past.

The outcomes of turning horses away were described with both positive and negative inflection. The majority of positive comments regarding reducing the structured care for the horse were focused around protecting human health. Industry professionals and owners with horses at livery discussed the benefits of turning away to reduce footfall on the premises which, in turn, enabled social distancing to be employed. Furthermore, the proprietor's ability to protect staff, and to disinfect surfaces and shared equipment was also seen to be improved where the workload was reduced through turning horses out. All groups recognized the reduced risk of handler injury as a beneficial outcome of turning horses away over confining them to stables, and the farrier group observed improved soundness in their client's horses, which they attributed to a more varied, or reduced workload. Finally, relieving boredom, increasing horse-horse socialisation, and promoting movement were seen as potential positive effects of increasing time at pasture.

Conversely, an increased risk of equine obesity, concern over laminitis due to unrestricted access to grazing, deterioration in horse behaviour and the potential for neglect were highlighted by interviewees as negative consequences of turning horses away. Restrictions to the ability to control the care for their horse were seen to be detrimental to horse owner mental health which, as previously stated, is out with the scope of the present analysis. It should be noted that negative mental health consequences were perceived predominantly in horse owners with animals kept away from their homes. Divergence in the experience of livery yard and home-based groups of horse owners was prominent in regard to public health measure implementation.

**Changes in veterinary and hoof care routine.**   Veterinarians noted an increased workload throughout lockdown, largely due to furlough of practice staff, increased telemedicine, and implementation of biosecurity related practices. The consensus amongst veterinarians was that the management of laminitis cases was not significantly impacted by practice changes. This was due to the emergency status of suspected laminitis cases, and visits to such cases were carried out as normal. However, veterinarians did comment that the less frequent interaction with horses during the early phase of lockdown accentuated cases of weight-gain and obesity. Farriers also generally noted changes in working patterns; generally indicating that workloads had increased or remained steady. The increased workload highlighted by veterinarians and farriers at the point of interview was notably due to the "backlog" (quote from F4) of work corresponding with an increased urgent need of care following initial efforts to reduce visiting professionals on yards during the early stages of lockdown.

An increase in business was often referred to when discussing farriers who had been self-isolating. The suggestion that working farriers may have intervened to take on the clients of those who reduced working hours during lockdown was implied, although this was only explicitly stated in a single farrier interview. Three of the four farriers interviewed had changed their "non-essential" clients' trimming and shoeing cycles during lockdown, most commonly by adding two to four weeks onto intervals between their regular visits. One farrier noticed a difference in the hoof growth due to this extension:

> "I was certainly surprised at how much extending the time between visits did in a couple of cases. One in particular, an ex-laminitic you could say, I was surprised at how much the growth had spread in that time."
>
> –F3.

There was an indication that extended time between trimming had been a contributing factor to laminitis development in two distinct cases discussed by one veterinarian and one owner. In both instances, a combination of factors were considered in the discussion about these cases:

> "my Highland developed a very severe bout of laminitis. . . I think [the laminitic episode] was [triggered by] a combination of things. . .The farrier was due just at the beginning of lockdown, and so their trims were put off. . .in-line with not doing any unnecessary shoeing. So, they weren't trimmed. And so, their feet were long for them, longer than they would usually be"
>
> –HH6.

Farriers indicated that the decision to return to normal shoeing and trimming cycles was prompted by the backlog of work which built up during lockdown- rather than in response to the public health risk being reduced.

Similarly, not all veterinarians expressed confidence in reinstating routine visits after lockdown. Two interviewees within the veterinary group indicated a sense that veterinary

authorities were abrogating decision-making responsibility to the veterinarian without clear indication that the risk of contracting and spreading the virus was reduced. In contrast, others felt that the Royal College of Veterinary Surgeons (RCVS) and the British Veterinary Association (BVA) issued ample guidance, and that following standard biosecurity protocols was sufficient for their practice:

> "I wasn't really studying every bit of guidance I was getting. I was just following the simple biosecurity measures that we are always aware of; I have no doubt that the Royal College and BVA have been setting guidance"

> –V2.

It may be relevant to note that those vets in mixed practice with frequent involvement with farm animal species tended to discuss biosecurity measures as being straightforward to implement on equine premises.

The equine veterinarian group showed the greatest level of concern over the possibility of spreading the virus to others and drew from experience and training in dealing with more common zoonoses, referencing both mRSA, and other infectious animal diseases such as foot and mouth disease. Vets suggested that they were, "just getting on with it", with regards to restarting activities at a time that may not have felt safe in a biosecurity sense. Whilst veterinarians were the among the most likely group to discuss the need to adhere to "essential practices only", in keeping with the public health advice issued by the government, they were the only group to discuss the health implications that visiting multiple premises in a single day could pose to themselves and vulnerable clients. Welfare centre managers expressed respect and acceptance regarding the practices that their veterinarians would and would not continue to perform during lockdown. However, there was tension between the horse owner demand for routine treatments and the vets' need to adhere to public health guidance. Horse owners tended towards frustration and fear that their horses' routine veterinary care, specifically vaccinations and dental check-ups, were not upheld:

> "So, with battle from the vets, I had to ask a number of times- I need you to come out and do vaccinations, you're coming out anyway. I was able to work with them eventually, through coordinating with the other horses that needed to be seen. Then by that time, he needed his teeth done, he was overdue, and the poor vets eventually said we will just do it, to save arguing with me really."

> –HL1.

Despite several concerns over routine work, no interviewees provided accounts or discussed experiences of difficulties accessing emergency veterinary or farrier care. On the contrary, horse owners and welfare centre managers frequently expressed gratitude for rapid and effective responses from these two practitioners in response to laminitis (1), euthanasia (2), and ongoing injuries (2).

> "it was actually very slick. The vets had a plan and had clearly thought about it. . .When I phoned the vet, I said it was an emergency and they were out within an hour which was brilliant"

> –HH1.

**Restricted owner-driven control over routine.** In the cases of owners with horses at livery, a loss of control over horse's routine was a prominent facet of conversation, although this

was not always discussed negatively. Horses' routines were subject to change only where they were housed on external premises, e.g. livery yards. No owners with their animals kept at home experienced alterations to their horse's routine; conversely this group noted a greater level of control over their animals' care. The degree of control that owners with horses at livery retained was dependent upon the severity of public health measures employed by proprietors. Livery yard owner approaches to reducing footfall and promote health on their premises fell under four categories: a total ban, a time-slot approach, hygiene measures only and little to no change.

The overall perception of the complete closure approach was negative. This was mostly due to a perceived lack of employees to maintain the horse's routine, thus risking a change in routine that may negatively impact the horse's health. Secondly, closure of yards was negatively perceived to result in a reduction in the number of times horses were being checked over (for measures of health) each day. Although veterinarians and farriers saw yard closures as responsible practice in terms of biosecurity, both groups expressed concern over animal health where significant routine changes were made. An account of such a scenario was described by an interviewee in the farrier group, who recounted weight loss as a concern in one case of yard closure:

> "on that yard, the horses weren't getting as well looked after. They weren't getting groomed, their extra bit of hay, and a lot of them lost a lot of weight, some were getting ulcers, just because it was too big a change to the routine. They just couldn't cope with it."

> –F4.

On the opposite end of the spectrum, veterinarians commented upon yard closures and reduced owner contact as disrupting weight management routines which had been effective previous to the lockdown:

> "some horses I have seen post lock down, I have never seen them so fat, ever. Horses I have known for 6–7 years. I have said to the owners, "oh my god what has happened? They have put on so much weight!". And they have said, "well for two months I haven't been allowed up to the yard, or I have been allowed up to literally come and pat it and check it over in the field"."

> –V3.

Allowing owners access to their horse on a rota basis was positively viewed where the horse's routine was maintained. Owners who's horses were managed with this approach appeared satisfied that they had maintained control over their horses routine, but did also highlight that slot booking systems needed to be flexible, provide at least 2 hours per person, and take into account time for riding, to be favourable. There were two direct accounts of owners at livery finding their horses' routines negatively impacted; in one case the owner was a veterinarian working unpredictable patterns, who found it difficult to, "commit to a single slot" (quote from HL5). In the second account, the owner disliked providing their horse with only two portions of forage per day, whereas normally they would visit their horse repeatedly to provide smaller portions as a dietary strategy to minimize intake and manage the horse's equine metabolic syndrome.

The final two approaches taken were to employ handwashing and disinfecting rules for those visiting without restrictions to number or length of visits, or to allow unlimited access to the yard and horse, with minimal biosecurity rules in place.

In an account from an owner with a horse at livery where minimal rules or restrictions had been employed, laminitis was seen as something which had become less of a priority during the pandemic.

> "people are busier managing their own horses and farriers and vets, those that should have been checked more regularly are not being checked. . .the laminitics are not a priority as much as they have been in previous years."

> –HL2.

## Outcomes of minimizing the risk of potential injury

Industry specific guidance discouraged horse owners from participating in activities which involved increasing the risk of injury. This information was disseminated through equestrian organizations and was issued to horse owners in various forms. Some authorities advised owners to reduce the range of activities they engaged in with their horses, whilst others advised that they stop riding completely.

**Horse riding "ban".**   Government guidelines for minimising the pressure on the NHS recommended that individuals did not engage in risk sports, such as horse riding. This guidance was reiterated by some, but not all, equestrian authorities which left some horse owners confused over the precise rules regarding exercising horses. Some interpreted equine specific guidance as a ban on riding. Others saw riding as necessary for preventing obesity and minimizing the risk of laminitis, as well as being justified within the Government's allowance of leaving your home for a single exercise activity each day. The result was a divide in the community between those who continued to ride, and those who did not, summarised by an account described by a horse owner below:

> "I was riding one day, and this lady started shouting at me and said that I wasn't allowed to be riding. She had her horse at livery, and they weren't allowing people to ride their horses, and she said, you have to look at the BHS, it is illegal to be riding your horse right now. So, I went and looked at the BHS and it said it is kind of up to you whether you want to ride your horse or not."

> –HL4.

The interviewee quoted above indicated embarrassment and frustration at the lack of distinct guidance that applied to their circumstances specifically- their horse was prone to laminitis and obesity and they were using exercise as a preventative method. One horse owner stopped riding immediately after seeing this guidance, and their Connemara pony developed laminitis soon there-after. Upon reflection, this owner stood by their decision, and was firmly of the belief that the risk of injury and of increasing pressure on the NHS, was such that it was not worth riding to potentially reduce the risk of laminitis. One welfare centre manager summarised the concept of prioritising human health over preventative health care in the equine, saying:

> "You know, you are trying to save your own life first. Not the ponies because they are a bit more resilient that we give them credit for"

> –WCM1.

Equine veterinarians tended to support restrictions upon riding and expected owners to alter their riding priorities during lockdown, although the lack of exercise that many horses

would receive raised worries from veterinarians and welfare centre managers in terms of obesity in particular. Equine veterinarian V5 commented that:

> "the lack of exercise for some horses did kind of worry me–although I agreed with it–was that places were stopping people from riding or exercising their horses. I agreed with it in terms of the pandemic, and to protect the NHS, but I worried about it in terms of the obese ponies that were probably just out on grass now."

> - V5.

**Reducing horse handling.**   Welfare centre managers highlighted the risk of injury associated with routine horse handling practices, particularly in the context of rehabilitating or retraining the horses on their premises. There was a significant aversion to undertaking practices which would place unnecessary pressure on the NHS:

> "as a manager, I didn't want to be the person driving someone to A&E with a broken finger, or a broken toe. Because we know that the serious injuries happen and of course you have to go to hospital, but the stupid, petty injuries also happen every day, and you still have to go to hospital. So, we elected to take all of our horses out of work, and to provide basic, primary care only. And that lasted for a period of about 4 weeks"

> - WCM 4.

For the horse owners interviewed, the risk of handling was acceptable. As a measure to manage weight gain, some owners opted to increase the amount of "in-hand" work (exercising and training horses from the ground) as an alternative to riding during the lockdown period. This approach to exercising the horse was endorsed by the veterinarian group as a potential intervention for preventing weight gain where riding was not suitable. The general consensus from horse owners was that their horses would become unmanageable if they were not stimulated with an exercise routine of some form. The farrier group noted a significant deterioration in the behaviour of horses they attended and attributed this to the reduced handling that some horses may have been receiving.

Welfare centre managers were often eager to resume retraining activities with horses undergoing rehabilitation, however this was more for the purpose of rehoming- this group did not note the development of new negative behaviours in horses under their care, perhaps due to the routine nature of handling horses with less training. With regards to laminitis cases, WCM1 commented that despite high levels of demand on rehoming centres to house laminitic ponies, the realities of what was achievable in the midst of the pandemic made taking these ponies on an unrealistic solution. This was in part due to the high level of close contact between handlers and veterinarians, and the intensive management, required to house a laminitic horse or pony, which may not be in the best interest of the centre or the veterinarian in terms of maintaining safe hygiene practices during the pandemic.

> "in this pandemic situation, if we were to bring in laminitics, we would have to make a serious decision on- are we going to continue with this horse or pony or not? And with so many chapping at the doors, more likely than not, we wouldn't have to luxury of pulling everyone through it"

> –WCM1.

## Discussion

The present study utilized qualitative methods to contextualize the secondary impact that the COVID-19 pandemic had upon the welfare of horses and ponies at risk of obesity and laminitis, from the perspectives of key industry stakeholders. Results showed that guideline interpretation, maintaining equine management whilst implementing public health measures, and minimizing the risk of injury were three lockdown-associated barriers to manging native-breed horses and ponies during the pandemic. The in-depth interviews conducted in the present study supplement the findings from national surveys [9] through providing a community level account of the drivers behind altered approaches to equine management. These interview data have enabled a deeper understanding of how stakeholders conceptualized the pandemic, the lockdown, and the associated guidelines, and how this translated into the care and management of native-breed horses and ponies at risk of obesity and laminitis.

### Challenges around guideline interpretation

Government advice during lockdown promoted the implementation of public health measures in shared spaces. Such practices are common in disease containment, and include actions which promote social distancing, isolation, quarantine and community containment [22]. On a national scale, individuals have continued to live under pandemic related restrictions and public health guidelines for almost one year (at the time of writing). Despite the continual evolution of advice over this time, the effectiveness of government guidelines to promote the pandemic response in the general public has been questioned [23]. The Behavioural Insights Team (Bi Team), a government-based policy institute which utilizes socio-economical insights to develop effective public and veterinary health interventions, surveyed the general public in England to understand how accessible COVID-19 related guidelines were [24]. It was demonstrated whilst the sample population could largely identify their regions COVID alert level, fewer were familiar with more detailed rules regarding support bubbles. This suggests that messages sharing government recommendations may have been insufficiently detailed for the general public to develop a robust and in-depth understanding of acceptable practice during lockdown. On livery yards, the most common public health measures to be employed were found by Williams et al. (2020) to be the provision of hand-washing facilities and the implementation of a time-slot system for horse owners to visit their horses [9]. This is consistent with the findings in the present study. However, interviewees provided no evidence or awareness of receiving or being offered training in human biosecurity following the UK lockdown. Decision making by livery yard owners was therefore likely to have been informed by past experience, training prior to the COVID-19 pandemic, observation of other yards, or guidance issued by authoritative equestrian organizations (of which there are numerous [25].

Disparate messaging from multiple sources may have led to different approaches by yard owners in managing their facilities during lockdown, a phenomenon noted amongst equestrians by Schemann et al. [7] during the Australian equine influenza breakout in 2007. In addition, results from the current study suggest that extensive planning, comprehensive yard rules, and strict biosecurity protocols were more likely to be positively perceived by industry stakeholders. These findings are consistent with those of Spence et al. [26], who found that, with regard to exotic disease, horse owners were more likely to perceive others as responsible where they had employed measures to minimize the spread of disease. The reputation of equestrian businesses, as well as the health and safety of individuals who access those businesses, may be promoted through the provision of biosecurity training to the managers of equestrian establishments. Such training may encourage a single doctrine to be adopted across the industry,

thus improving the confidence of stakeholders in the decision making of local equestrian establishments.

The question arises, from where should horse managers have sought guidance to inform the public health measures they enacted on their yard? In the US, the initiative was taken by the, "eXtension", group, a collaborative effort between equine industry experts, to provide webinars, podcasts and infographics outlining best biosecurity practice for equine businesses [8]. Eighty percent of 138 survey respondents who had accessed eXtension's webinars on finances, biosecurity and contracts during the pandemic, said that they would utilize the information gleaned from the presentations. Taking direction from this approach, future efforts to define, condense and disseminate information regarding horse management during public health emergencies should ideally involve expert collaboration from equine veterinarians and experts in the fields of biosecurity, equine health and epidemiology. Fineberg (2016) [27], drawing from knowledge acquired from the 2009 H1N1 pandemic, foresaw that a lack of cohesive, succinct information could be a pitfall of large collaborations which seek to address specific pandemic related questions. This highlights that strategies used for information dissemination can be as important as the information itself. To support future initiatives to share information with the entire equestrian industry, in-depth assessment of the information consumption by UK stakeholders should be performed to optimize pre-existing information sharing networks in the industry, as well as to develop new avenues of sharing.

Utilising the type of expert collaboration recommended above, the BFBA developed guidance for farriers through consultation with key veterinary, welfare and research authorities in the equestrian industry. The resulting guidance included a "traffic light system", which directed decisions regarding which horses warranted immediate hoof care [28]. This was disseminated to farriers through the "Forge and Farrier" website, an online resource developed for the purpose of sharing information with registered farriers and blacksmiths. Consistent messaging, and the appearance of a united approach by the farrier community to handling the lockdown were highlighted as practices which provided horse owners with confidence. Whilst this approach to information sharing may be the reason for farriers perceived success in sharing information, it may also be due to use of the "traffic light system" itself. A similar strategy was recently developed by BEVA in collaboration with the Bi Team, who implemented a colour-coded monitoring system to facilitate communication between vets and horse owners surrounding the awareness of equine obesity [29, 30].

Efforts focused towards the production of synchronized guidance for horse owners, equestrian establishment owners and managers, and other industry stakeholders, could improve the response and adherence to guidelines regarding public health in the future.

## The implications of public health measures on routine equine care

The current study identified that increasing time at grass for horses was a practice employed by equestrian establishments during lockdown. This led to significant concern amongst equine veterinarians, horse owners and welfare centre managers over the repercussions that extended periods of uncontrolled grazing could have on equine welfare, as well as concerns from a welfare perspective around the increased demand on rehoming as an inevitable result of the pandemic [31, 32]. Increasing turnout promotes time spent foraging, whilst reducing negative stereotypic behaviours such as weaving and crib-biting [33]. However, increasing access to pasture, changes in pasture consumption and seasonal fluctuations in carbohydrate content of forage, have been linked with obesity [34–37], endocrinopathic laminitis [10, 14, 36–39] and gastrointestinal disease [40]. Lockdown measures were announced on the 23rd of March–a time of year at which grass growth can be subject to conditions that promote the accumulation

of non-structural carbohydrates in the plant [41, 42], and therefore increased grazing would have likely posed a greater risk to those animals at risk of laminitis [43, 44].

In addition to increasing time at pasture, the recommendations often suggested that owners "rough off", horses [45]. Like turning away, roughing off is the process of gradually transitioning a horse into a more naturally maintained state, reducing structured care and exposing the animal to increased time at pasture in order for it to build tolerance to living outdoors. This may have effectively reduced the time horse owners were required to spend in a public space, thus addressing human health concerns relating to social distancing. This advice was often accompanied by a cautionary note to consider the requirements of horses at risk of laminitis and obesity. However, the present research has shown that this advice did not prevent these conditions from occurring in the animals under the care or supervision of the interviewees. This may suggest that, despite being aware of risk factors for obesity and laminitis, horse owners were not adequately informed of, or could not effectively implement practical management-based precautions which should be enacted to minimise this risk. Developing alternative strategies to turning away and roughing off that address public health requirements, as well as disseminating the risk-limiting advice [46–48] to owners of obesity and laminitis susceptible animals may be beneficial for the development of future horse care -related policies.

Given the relationship shown between turning horses away and the furlough of key equestrian staff, it may also be beneficial to consider the benefit that emergency legislation and funding to support key workers involved in the care of equids at risk of laminitis would have during emergency scenarios. Funding during the pandemic was identified as the most frequently cited concern of third sector organisations in a survey conducted by the Scottish Government [49]. Financial aid to protect those most in need during the COVID-19 pandemic was actioned in Scotland, with support services for key vulnerable groups receiving part of the Scottish Government's £350 million Communities Funding reserve [50]. In the equestrian industry, The Petplan Charitable Trust (PPCT), World Horse Welfare and National Equine Welfare Council collaborated to offer grants of £5000 for organisations involved in the rescue and rehoming of equines [51], and support for small equine businesses was available from the Government [52]. Considering the cost of upkeep of horses, estimated at £3,105 per horse per year in the UK [53], additional costs related with the diagnosis, treatment and prevention of laminitis and obesity could make caring for a laminitic horse an unfeasible expense for some. Thus, the allocation of emergency funding for the costs associated with laminitis care may help to support horses, equestrian businesses and workers in the equestrian industry in the future.

As was seen in human medicine and small animal veterinary practice [54], equine veterinary practitioners were required to rapidly adapt to tele-communication-based practice models as the pandemic progressed into lockdown. To reduce the staffing rates in individual practices, many vets were placed on furlough, reportedly affecting the mental health and well-being of BEVA members [55]. Retained veterinarians were advised by the RCVS to pause routine equine flu and tetanus vaccinations during, "full lockdown", which lasted for roughly seven weeks. Research has identified that horse owners look to their veterinarians for guidance, both in regard to aspects of equine care and welfare in the competition industry [56] and as a source of biosecurity information as illustrated with respect to the Hendra virus (a zoonotic virus which can infect humans and horses) in Australia [57]. This is interesting given the current study identified an apparent mistrust shown from owners towards veterinarians regarding the decision to pause vaccinations. In agreement with this, Williams et al. [9] identified concern amongst horse owners regarding their animals' immunity to equine influenza due to veterinary practices temporarily stopping routine vaccinations during lockdown. It is likely that the horse owners in the current study were concerned about the health implications of this decision and in terms of their eligibility to compete in affiliated events that require their

animals to possess an up-to-date vaccination schedule. The health implications of this decision could have been communicated by veterinarians to owners in an attempt to address these concerns.

In addition to veterinarians, farriers also appear to be a trusted reference for the equine community. In a survey of UK horse owners, Thirkell and Hyland (2017) [58] found that 73% of respondents trusted their farrier's knowledge and ability, "completely". This trust in ability may translate into trust regarding other aspects of equine care [59], including public health during the pandemic. The approach to limiting non-essential work practices for farriers by extending the time between trimming was largely accepted by horse owners in the current study, although, unlike pausing vaccinations, interviewees linked this practice to negative health outcomes for animals with active or historical laminitis. Pollard et al. (2019) [37], found that horses with active laminitic episodes were more likely to have been in trimming or shoe-ing cycles of intervals of more than 8 weeks apart, whilst Hockenhull and Creighton (2010) [60] found shoeing / trimming intervals of greater than 7 weeks to be associated with discom-fort behaviour in horses. The current study provided accounts of native ponies that had experi-enced previous episodes of laminitis being placed onto extended trimming cycles, which may suggest a lesser degree of concern for previously laminitic horses despite their increased risk [37]. It is possible that extending the time between farrier visits, as directed by the BFBA, could exacerbate the risk of laminitis in predisposed animals [61]. It is, however, important to note that horses with laminitis were all considered to be in the "crucial hoof care" category by those interviewed in the present study. Further work may be warranted to identify specific condi-tions, and "types" of horses that were perceived by farriers to warrant more frequent visits, whilst recommendations for hoof care during future emergencies should draw attention to the importance of maintaining routine hoof care to those animals deemed to be at greater risk of developing laminitis.

## Restricted owner-driven control over routine

In the current study, restricting access to premises was seen to limit the control that owners had over their horse's routine. However, this appeared to be dependent upon the degree of co-operation with the owner that the proprietor was willing to undertake. Schemann et al. (2012) [7], in regard to the equine influenza outbreak, revealed that horse owners and yard proprie-tors were more likely to view on-farm biosecurity measures as effective when the interviewee had not experienced the disease in their own horses. This is similar to the findings of qualita-tive investigations which have shown that those who perceive risk as high are more likely to implement mitigation measures [62, 63]. This phenomenon bears resemblance to the indiffer-ence that some interviewees of the present study expressed when discussing SARS CoV-19 infection itself, particularly where they had little or no connection to cases of infection. Inter-estingly, the present interviews were conducted prior to Aberdeen itself being placed into a "local lockdown" on the 5th of August, after an outbreak was traced to the city. Further inter-views to measure how attitudes have changed in light of the "closeness" of the virus to the Aberdeenshire region, would inform of the influence that the sense of being sheltered from the pandemic had upon the acceptance of more stringent public health measures on yards.

Veterinarians were the only group in the present study to explicitly highlight the risk of spreading the SARS CoV-2 virus when visiting multiple sites each day. Performing non-essen-tial procedures during lockdown whilst employing strict biosecurity measures, such as main-taining social distancing, was an apparent point of contention for some veterinarians interviewed. The risk to human health that close person-to-person contact during yard visits posed was generally perceived as more significant than the low level of short-term benefit that

routine treatments offered the animal. Similarly, questions over the safety of farmers and veterinarians during routine bovine tuberculosis testing were raised in the Veterinary Record [64], highlighting the need for careful consultation with practicing veterinarians when developing guidance to ensure practicality of human biosecurity measures in real life scenarios.

### Outcomes of minimizing the risk of potential injury

Advice was circulated from equine authorities which recommended that horse riders stop riding during lockdown to reduce the risk of unnecessary human injury, and therefore reduce pressure on the National Health Service (NHS) [45]. Response to this guidance divided stakeholders into those who wanted to continue riding to manage their horse's weight, and those who regarded horse riding as posing an unacceptable risk to human health.

There is a robust pool of evidence in humans which demonstrates the positive influence of exercise on obesity and insulin resistance [65]. In the equine, however, results from research assessing the role of exercise in managing obesity and insulin resistance are variable, and there are limited investigations into how exercise affects the metabolism of native breed horses and ponies specifically. These breed-types are thought to have evolved to extract maximal nutrients from their feed in order to survive their native environments [66], suggesting an adapted metabolism which tends toward energy conservation. Exercise has been shown to improve insulin regulation in the horse through promoting insulin sensitivity and enhancing pancreatic beta cell function [67, 68] and was also demonstrated by Menzies-Gow et al. (2014) [69] to significantly reduce markers of inflammation in the plasma of previously laminitic ponies. However, exercise alone may not be sufficient to promote weight loss in insulin dysregulated animals [70]. A combination of exercise and dietary restriction has been shown to reduce basal insulin concentrations and improve insulin sensitivity, however this regimen did not increase weight loss [71]. It would therefore, be advantageous to encourage the owners of obese, or insulin dysregulated horses to practice alternative exercise methods during lockdown situations that do not involve the same level of risk to the handler. Guidance on the subject of exercising horses might mitigate the impact of riding restrictions upon the welfare of animals, limit the risk to human health associated with riding, and provide owners with enriching activities to engage in with their animals during a lockdown.

In this regard, the number of self-reported incidents recorded by the BHS Safety department through the horse incident reporting site show that, between the months of March and August 2020, reporting of equine related incidents was reduced in comparison to the same time frame in 2019 [72]. The trend in the data (Fig 2) indicates a rise in reported incidents between 2017 to 2019, followed by a 1.5-fold decrease in reports in 2020. There are multiple factors that this reduction could be attributed to, and the reliability of self-reported horse-related injuries is unknown. As such, these figures should be interpreted with caution, and distinct conclusions cannot be drawn as to whether reduced riding caused a reduction in equine related injuries. More comprehensive data could reveal the true impact of reduced riding upon injury rates on hospital admissions and may provide evidence that could drive future recommendations regarding horse riding during public health emergencies.

### Limitations

The methodological approach utilized was nested in grounded theory, with an influence of interpretive phenomenology in its understanding that to translate lived experience into an explanation, the experience of the researcher inevitably influences the final report [73]. This qualitative methodology is set within the hermeneutic understanding that researchers take a degree of interpretative license during the evolution of study findings as they develop from a

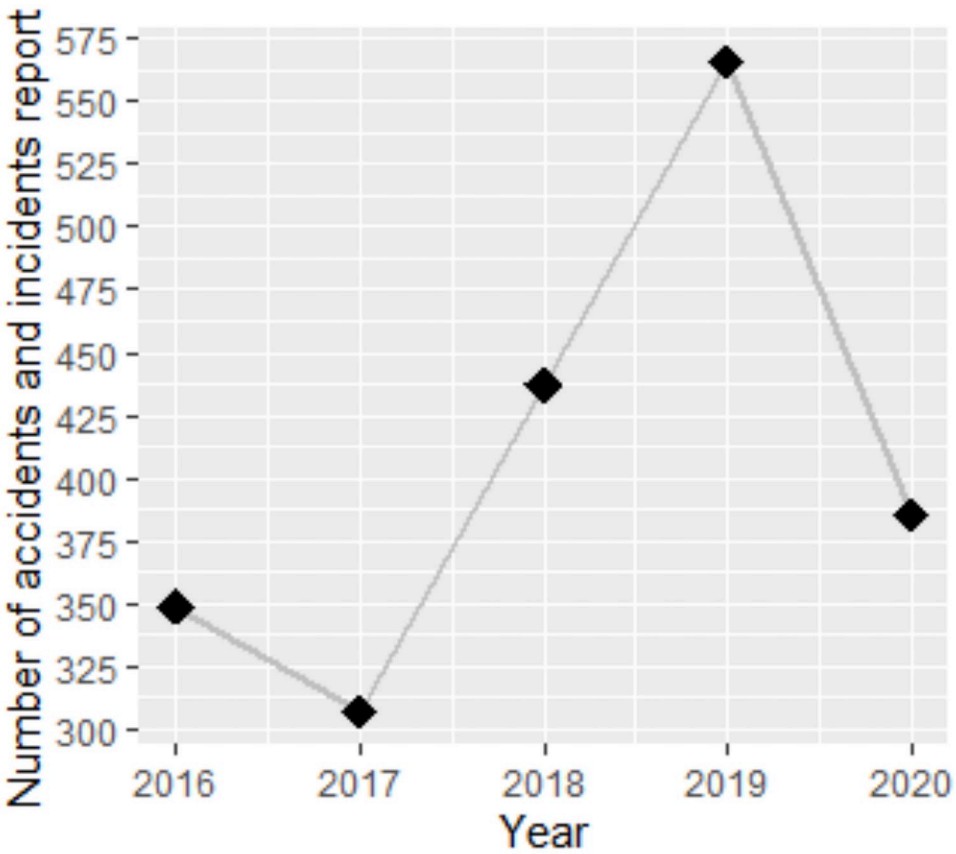

**Fig 2. Count of self-reported horse related injuries submitted to the BHS Incident reporting website between March and August across a 5-year timeframe.** The number of self-reported incidents and injuries relating to horses between the months of March and August across a five-year time period between 2016 and 2020. The figures shown are those submitted to the BHS incident reporting site [72].

transcript of lived experience, to a description of that experience filtered through the researcher's bias [74]. The research team had professional, and personal, experience of horse ownership and first-hand knowledge of the potential impacts of the pandemic upon equine laminitis management. Given their involvement in the industry, the primary researcher was casually acquainted with some of the interviewees through horse ownership. In the given circumstances, a working understanding of the equestrian industry was beneficial, if not necessary, to obtain specific insights that addressed the research questions of the study. It is accepted that, as a result, the findings of this study are a subjective product of the interpretation of interviews and transcripts through the perceptions of the author.

This research was subject to the inherent limitation of qualitative methodologies; specifically, the reduced applicability of the research findings to the wider equine stakeholder population. This is due to the nature of the purposeful, heterogenous sampling approach which enabled detailed insight into the subject area of native horse management and welfare concerns during the pandemic, but sacrificed the applicability captured through random sampling methods. Applicability of the findings may also be limited by the inclusion of two respondents (WCM3 and WCM4) who were based outside of the Aberdeenshire region. It is possible that the views and experiences of these two participants may not be comparable to those of welfare

managers based in the Aberdeenshire locale, but their inclusion was deemed necessary to deepen insights into the challenges faced by equine welfare centres.

Questions around the socioeconomic and educational backgrounds of the participants were not included in the interviews, meaning that differences in the unique impact of the pandemic between subpopulations from different backgrounds cannot be inferred from the results of the present study. However, the questions were associated with native horse management and therefore the respondent answers may be relevant across the UK at least. Given the reach of the pandemic across the world, and the psychological impacts highlighted within the study, certain aspects of the study may still be applicable to wider groups. Finally, although the target of recruiting 5 participants per stakeholder group fell one participant short for two groups, the data and resultant coding generated from these groups was such that this smaller sample size was considered to have captured sufficient information to assume thematic saturation.

As interviews were conducted by telephone, capturing the nuances of communication during face-to-face interview techniques was not possible. Despite this limitation, the use of telephone interviews allowed extensive notes to be taken during conversations without disrupting the fluency of the interview.

## Conclusions

The current study identified four key themes that emerged from stakeholder interviews: challenges around guideline interpretation, implications of public health measures on routine equine care, outcomes of minimizing risk of physical injury, and negative impacts of the pandemic on mental health. The detailed accounts achieved through qualitative methodology revealed that the policies implemented to protect public health during the pandemic had secondary impacts upon the welfare of native-breed horses and ponies at risk for obesity and laminitis. Ensuring that the exogenous risk factors for obesity and laminitis are highlighted when issuing equine related guidance could improve the welfare of susceptible horses. Furthermore, a collaborative, multi-industry approach to developing and issuing equine specific advice has the potential to improve continuity in the measures adopted across individual equine establishments. Lessons for policy makers should include an appreciation of the interaction between the time of year and equine welfare during future lockdown events. Finally, the qualitative methodology adopted in this study support its use in highlighting novel phenomena occurring at the community level which other forms of research may overlook. As such, repeated evaluations of the ongoing impacts of the pandemic upon equine welfare will improve understanding of policy implications, and policy development in the future. It may be beneficial to consider the present findings when developing guidelines to protect public health in the equestrian industry during future public emergency scenarios.

## Supporting information

**S1 Table. Inclusion and exclusion criteria.** The inclusion and exclusion criteria used for sampling appropriate sub-populations to gain insight into laminitis and obesity concerns during the COVID-19 pandemic.
(DOCX)

**S2 Table. Classification of interviewees, additional details and length of interviews.** Units of study. The characteristics of each participant are presented in S1 Table. This concise and purposefully selected sample was agreed upon as providing a wealth of experience on the subject of managing laminitis, and as being directly impacted by the pandemic.
(DOCX)

**S3 Table. An example of the coding process used to determine appropriate themes.** Examples of direct interview text and the categorisation of data. Meaning units were condensed, and theoretical codes assigned, and the underlying meaning of the quote was interpreted utilising annotations taken during interviews. Sub-themes and themes then emerged as the frequency and importance of concepts arose.
(DOCX)

**S1 Fig. Categorical codes, sub-themes and higher-order themes generated in the current study.**
(TIF)

**S1 File. Anonymised interview transcripts.**
(PDF)

## Acknowledgments

We wish to extend our gratitude to the local horse owners, veterinarians, farriers and welfare centre managers who volunteered their time to take part in this research. Our thanks also to Dr Charlotte Maltin for supporting recruitment for the study and to World Horse Welfare for their continued interest in the key welfare issues addressed in the present study.

## Author Contributions

**Conceptualization:** Ashley B. Ward, Kate Stephen, Caroline McGregor Argo, Patricia A. Harris, Christine A. Watson, Madalina Neacsu, Wendy Russell, Dai H. Grove-White.

**Data curation:** Ashley B. Ward.

**Formal analysis:** Ashley B. Ward, Kate Stephen, Patricia A. Harris, Christine A. Watson, Madalina Neacsu, Philippa K. Morrison.

**Funding acquisition:** Philippa K. Morrison.

**Investigation:** Ashley B. Ward, Kate Stephen, Patricia A. Harris, Christine A. Watson, Madalina Neacsu, Wendy Russell, Dai H. Grove-White.

**Methodology:** Ashley B. Ward, Kate Stephen, Caroline McGregor Argo, Philippa K. Morrison.

**Project administration:** Ashley B. Ward, Philippa K. Morrison.

**Supervision:** Kate Stephen, Caroline McGregor Argo, Patricia A. Harris, Christine A. Watson, Madalina Neacsu, Wendy Russell, Dai H. Grove-White, Philippa K. Morrison.

**Validation:** Kate Stephen.

**Writing – original draft:** Ashley B. Ward.

**Writing – review & editing:** Ashley B. Ward, Kate Stephen, Caroline McGregor Argo, Patricia A. Harris, Christine A. Watson, Madalina Neacsu, Wendy Russell, Dai H. Grove-White, Philippa K. Morrison.

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
