## [Decision Letter · Decision Letter 0]

1 Mar 2021

PONE-D-20-38153

COVID-19 impacts equine welfare: policy implications for laminitis and obesity

PLOS ONE

Dear Dr. Morrison,

Thank you for submitting your manuscript to PLOS ONE. After careful consideration, we feel that it has merit but does not fully meet PLOS ONE’s publication criteria as it currently stands. Therefore, we invite you to submit a revised version of the manuscript that addresses the points raised during the review process.

As you see, the reviewers' evaluations are mostly positive. So is mine. At the same time, however, the Reviewers also raised quite a lot of remarks and suggestions. Hence, the manuscript needs some revision before it might be accepted. First of all, you should reduce the text to make it more concise and specific to obesity and laminitis of the horses.  Also, the methods require clarification. You should provide us with further specific details about your sample, such as the horse owner's information, etc. (see more in the comments of Reviewers 2 and 3, and also the point 1 to 14 by Reviewer 1). 

All reviewers provided numerous useful detailed suggestions. Try to consider them if you decided to revise your manuscript. I will ask them again to say how you coped with their comments. I believe there is a strong potential for the study to be an excellent, interesting paper. 

We look forward to receiving your revised manuscript.

Kind regards,

Ludek Bartos

Academic Editor

PLOS ONE

Journal Requirements:

2. In the Methods section, please provide details regarding how verbal consent was documented.

4. Thank you for stating the following in the Financial Disclosure section:

'This study was funded by Mars Petcare and is part of a PhD studentship funded by the Scottish Funding Council Research Excellence Grant (REG). Authors WR and MN receive salary support from the Rural and Environment Science and Analytical Services Division (RESAS).

PH was involved in study design, data interpretation, and manuscript preparation.'

We note that you received funding from a commercial source: Mars Petcare

Reviewers' comments:

Reviewer's Responses to Questions

**Comments to the Author**

1. Is the manuscript technically sound, and do the data support the conclusions?

Reviewer #1: Yes

Reviewer #2: Yes

Reviewer #3: Partly

2. Has the statistical analysis been performed appropriately and rigorously? 

Reviewer #1: N/A

Reviewer #2: N/A

Reviewer #3: Yes

3. Have the authors made all data underlying the findings in their manuscript fully available?

Reviewer #1: Yes

Reviewer #2: Yes

Reviewer #3: No

4. Is the manuscript presented in an intelligible fashion and written in standard English?

Reviewer #1: Yes

Reviewer #2: Yes

Reviewer #3: Yes

5. Review Comments to the Author

Reviewer #1: This is a wonderful paper, and it was a pleasure to have it drop into my inbox on a Monday morning. It is well-written and structured, takes a thoughtful and careful approach to considering how public health and obesity/laminitis are linked in the Covid pandemic, and is methodologically sound. I particularly liked the holistic approach to examining the issue, bringing together multiple stakeholders but particularly farriers, whose views are often overlooked, though they are absolutely key in obesity and laminitis management. I see this is the first author's first paper from her PhD, and I therefore particularly congratulate her on producing such an excellent piece of work so early in the process.

I recommend that is is accepted - I have some minor thoughts and comments. Often these are to do with tightening things up - you're trying to cover an awful lot because your data clearly is very wide-reaching, but your paper is focussed on obesity and laminitis, so just need to continually reflect that and not get side-tracked (it's already quite long).

1) the list of themes - I think these are interesting and well presented. however, in line with GT approach I wondered if you'd considered how they connect to one another.

2) in section 1.1. you talk about the horse owners not having coherent advice from any one place, but not about professionals - could add a little info here about whether vets, farriers and welfare staff had coherent info (or just clarify that you discuss this later)

3). Section 1.2 - you have some quotes within the text (such as "sing from the same hymn sheet") but it's not clear if they're actually from participants? If so, would be nice to just add the info (e.g. " quote from Vet 1" or whatever)

4) Section 1.2 - in your initial list of codes this section is called "positive IMPACT" and in the section heading later it is called "positive EXAMPLES". As "examples" I am not convinced it is really a sub-theme, but when called positive impact it makes more sense.

5) Section 1.2, you mention the farrier traffic light scheme a few times - need to explain what this is

6) All of section 1 - you don't mention laminitis/obesity here, which is fine as I know it sets the background for the following sections. However, perhaps you could signpost this; for example before going into detail on the results you could say "firstly we will describe the way in which the guidelines affected the equestrian community generally, before considering how these impacted on equine management in relation to laminitis and obesity" (or somesuch) - just the first time I read it, I had to remind myself where I was in the paper, that's all.

7) Section 2.2. first you mention an increase in work, but next that there was LONGER between trimming cycles, so this is a bit confusing (esp as the section is presented as if reducing visits from professionals was one thing yards implemented to reduce disease risk, but then start the section by saying they actually did more work). I think this could just be re-ordered and have a bit more explanation to clarify.

8) Section 2.2. - this section could be linked a bit more closely to laminitis and obesity I think? It's only mentioned in the middle

9) section 2.3. "restricted owner control..." - v interesting section, just not convinced the section name actually covers its contents! It partially does, but you actually focus more on yard visits/bioscurity than what the owner can or cannot do with their horse. I can see why biosecurity is relevant here, but wonder if a) you could reduce the amount you say about it so you only talk about it in reference to owner restricted control on their actual hrosecare and not more generally, and/or b) talk about yard restrictions/increasing biosecurity elsewhere

10) Line 513 authority's should be authorities

11) section 3 - Need to explain why authorities suggested reducing activity/risk of injury, and perhaps in the intro say why this is relevant to obesity/laminitis

12) 3.1 and 3.2 link more clearly to obesity/laminitis

13) line 643-644 about traffic light scheme - it doesn't help owners MANAGE obesity at all, it just is supposed to help vets bring up the issue with owners (trying to help awareness)

14) line 703-705 - there must be a more recent UK example of owners looking to vets for advice? Just not sure a 1994 outbreak of a disease not v well known here is a good example.

I think that's it! Well done again.

Reviewer #2: This study explores the challenges that recent lockdown events have posed to equine welfare, specifically in laminitis and obesity susceptible animals. Opinions from a range of stakeholders have been considered and this range of perspectives have contributed to the themes identified. The study highlights important issues such as the need for clarity and consistency in guidance issued and will provide a useful point of reference when considering the potential impact of further public health measures on UK equine welfare.

I enjoyed reading the paper and think it raised some interesting points. Suggestions for minor revisions are below:

Introduction:

The introduction is cohesive and clearly states the aims of the project.

Methods:

In general the methods require some clarification.

It is stated that pre-defined questions and structured prompts are used, however in L149 you indicate that the structure and direction of interviews was determined by the participant. This seems like a contradiction.

With the wording of questions varying, would you not expect to see differences in responses as a result? Similarly, with an iterative process how do you expect that the inclusion of new topics may have influenced results?

It is stated that you used a ‘targeted direct approach’ to sample participants but later state that it was a ‘pragmatic approach involving convenience’. To me this seems more like an opportunity sampling strategy within your five pre-defined groups.

Further detail about your sample is needed. Please clarify whether all participants with horses at livery were from different livery yards. More information on your horse owner (livery and home) samples would be beneficial – for example whether all owned leisure horses or if some were sport/competition horses. ‘Horse owner’ can cover a wide range of people in terms of socioeconomic or educational backgrounds, it would be interesting to know whether all horse owners came from similar backgrounds or whether a variety of perspectives were included. If this data wasn’t collected, the point could be added to the study limitations. Also whether the livery yards that your sample used were similar in their size/structure/pricing.

In L148 could you add which type of average you calculated - mean, median etc.

It is mentioned that codes were based on ‘suitable categories’, could you expand on this? Was this based on topics that came up most frequently? Similarly, the sentence spanning L181-184 is not very clear, perhaps an example would help illustrate your process

Results:

There is an error in L208 – do you mean ‘outside of’?

In L239 a phrase is used in quote marks. Was this a direct quote from a participant (if so it would be useful to state a participant number) or a figure of speech from the author? The same is seen in L254 and L258.

I agree that it is important to distinguish between perceived and actual reasons for turning away.

In L395 furlough is mentioned regarding veterinarians, could you clarify here if you mean furlough of colleagues or the general public?

There is an error in L513 – authorities

Discussion:

In general the manuscript could be more concise, there is repetition of content in the results and discussion that may be able to be cut down.

The use of ‘mistrust’ in L701 is quite strong, perhaps concern is more appropriate.

You address the topic of subjectivity and generalisability in your limitations section nicely. However, I think that reference to your small sample size should be included here, as you stated that 5 participants were needed as a minimum for gaining sufficient information from a group, and in two of your five groups this figure was not reached.

Reviewer #3: An interesting study and good to see researchers determining the impact of the pandemic on equine welfare. Your work would add to the growing body of work in this area but currently it feels like a more general review of the impact of the pandemic on equine welfare is being shoehorned to focus on laminitis and obesity, and for me this detracts from the quality of your work and would suggest rebranding with a more general lens of the impact on ‘equine health and welfare’. I would also urge the authors to consider if a more local/ regional perspective would better represent their sample as this would provide a detailed community level response (as limited – 2 – not in a local radius).

Comments

Title

While obesity and laminitis were some key concerns arising from your interviews they were not the key focus of the interview approach taken and I feel your work is a more general review of the impact of the pandemic on equine welfare and your title should reflect this

Abstract

It would be beneficial to include a more general summary of the themes that arose from your analysis then to highlight laminitis etc as this would be more representative of the work undertaken

Line 30: there are a number of published studies which have considered the impact of Covid on horses which include welfare aspects, therefore suggest amending this statement to something like warrants exploration

Line 34: suggest amending to laminitis were:

Line 41: would be good to identify a context for the guidelines referred to e.g. management broadly within the pandemic or specifically for at risk horses / ponies

Introduction

Rationale for study provided complemented by background to the pandemic and its impact in horse owners.

Line 60: please remove comma after associated

Line 62: suggest inserting: horse owners after some

Materials and methods

You could present the methodological framework that underpins your analytical approach and interpretation more explicitly in the opening paragraph.

Please define your inclusion / exclusion criteria for potential participants; it would be good to align these to your 5 key stakeholder groups

Your sample is largely local (Aberdeenshire) – given the differences between UK regions with respect to lockdown and quarantine regulations, I would advise the authors to consider removing the Somerset and Blackpool participants to give your work a local or Scottish focus. If you elect not to then the potential limitations of the sampling strategy should be considered in your discussion.

Please include details of who conducted the interviews and outline their experience with this process. Did the interviewer take notes during the interview? If yes, please discuss the limitations of this approach.

Line 148 – 149: please edit sentence to enhance flow as it is a bit disjointed

Line 167: suggest phrasing as Supplementary file 1: Table S1

Line 178 it would be beneficial to reiterate the methodological framework applied across coding within this paragraph and to add citations to support the approach taken (refer to Braun and Clarke’s work)

Please can you clarify how many of the research team analysed the results and if triangulation occurred at this level or was a review of the summary table only

Did you engage in any stakeholder verification of the results?

Results

Detailed discussion of themes and sub themes presented; does overall feel a little bit negative and I wonder if there are some more positive aspects which you could highlight to balance the selected quotes more.

Line 191: I think identifying your 4 key themes here would be beneficial

Line 195: I would consider your presentation of the higher/ lower order themes as a list – I think a table or figure would enhance the impact of your work (and include theme 4); it may be nice to include some contextualisation to add flavour to the theme / sub themes presented as well

Line 208: please edit sentence suggest is out of the scope of the present article

Discussion

Interesting points debated but the discussion is a little lengthy and in parts a little repetitious to results; would suggest an edit to make more concise would be beneficial to increase impact

It would be of interest to evaluate if the approach to the pandemic perceived by equine stakeholders is unique or if similar patterns occurred across other groups e.g. small animal vets, dog owners, animal shelters, homeless shelters etc reflecting a more global pandemic phenomenon, within your discussion.

Line 569: please replace this with these

Line 647/8: there are examples of positive practices across different industries including equestrianism – may not be the headlines in research but they are there – also suggest reviewing some of the grey literature and industry sources in respect of this

Lines 655 / 663: please remove highlighted text

Conclusions

For me your conclusions should relate to the key findings of your work i.e. the 4 themes and subthemes as the headline and then you can link to laminitis and obesity – suggest reconsidering this section in line with broader feedback to shift the focus of the paper to broader equine health / welfare.

6. PLOS authors have the option to publish the peer review history of their article (what does this mean?). If published, this will include your full peer review and any attached files.

Reviewer #1: **Yes: **Tamzin furtado

Reviewer #2: No

Reviewer #3: No

---

## [Author Response · Author response to Decision Letter 0]

12 Apr 2021

Reviewer #1: This is a wonderful paper, and it was a pleasure to have it drop into my inbox on a Monday morning. It is well-written and structured, takes a thoughtful and careful approach to considering how public health and obesity/laminitis are linked in the Covid pandemic, and is methodologically sound. I particularly liked the holistic approach to examining the issue, bringing together multiple stakeholders but particularly farriers, whose views are often overlooked, though they are absolutely key in obesity and laminitis management. I see this is the first author's first paper from her PhD, and I therefore particularly congratulate her on producing such an excellent piece of work so early in the process.

I recommend that it is accepted - I have some minor thoughts and comments. Often these are to do with tightening things up - you're trying to cover an awful lot because your data clearly is very wide-reaching, but your paper is focussed on obesity and laminitis, so just need to continually reflect that and not get side-tracked (it's already quite long).

Author reply: Thank you very much for your kind and encouraging comments. We are delighted that this reviewer appreciated the importance of this study and greatly appreciate the comments which we have addressed individually below. We feel that the changes made in light of these comments have greatly improved the overall quality of this manuscript. Thank you. 

1) the list of themes - I think these are interesting and well presented. however, in line with GT approach I wondered if you'd considered how they connect to one another.

Author reply: Thank you for your comments. In response to your suggestion to consider connections between themes, the list of themes has been re-formatted to include greater detail and is now presented as a figure (Figure S1) to highlight interconnected concepts. For example, all of the primary themes contribute to the negative impact of the pandemic upon mental health, which has been demonstrated through connecting these aspects in the diagram. Furthermore, certain elements of the measures taken to minimise the presence on yard premises also contributed to decision making regarding whether or not to ride. Again, we have connected these points in the diagram to demonstrate this association. We hope that these amendments show our attention to your advice. 

2) in section 1.1. you talk about the horse owners not having coherent advice from any one place, but not about professionals - could add a little info here about whether vets, farriers and welfare staff had coherent info (or just clarify that you discuss this later)

Author reply: Thank you for this useful observation. We have added details here from the other groups’ experience to complete the picture. 

(L233): “In contrast, farriers found their guidance concise. All interviewees in this group identified governmental messaging and the British Farriers and Blacksmiths Association (BFBA) as the two sole sources of information regarding their working practices during the pandemic. Interestingly, the EWC group perceived the multiple sources of information as less of an issue, but as having the potential for positive impacts on information exchange. One welfare centre manager outlined the connections between the organisations and highlighted the veterinary associations as the welfare centre’s ultimate point of reference for information, saying: 

“you have NEWC, the National Equine Welfare Council, of which we are part of. That is basically equine charities which have all come together…. although we all work individually, we all know what each other are doing at one time. Obviously, the British Equestrian Federation, and BEVA, the veterinary advisory groups as well…our head office keeps abreast of exactly what they are going to issue, and we have to follow their guidelines.” – WCM1

Veterinarians consulted governmental advice, and that of the British Veterinary Association (BVA) and the British Equine Veterinary Association (BEVA) for their information. Although the number of sources wasn’t seen as overwhelming for this group, the slow release and lack of practical detail was highlighted as a challenge.”

3). Section 1.2 - you have some quotes within the text (such as "sing from the same hymn sheet") but it's not clear if they're actually from participants? If so, would be nice to just add the info (e.g. " quote from Vet 1" or whatever)

Author reply: Thank you. Details of the source of the quotes in section 1.2 have been added. 

[L257] ““glazed over” (quote from HL2)”

[L272] ““sing from the same hymn sheet” (quote from V4)”

[L276] ““you never know what you’re going into”, (quote from V4)”

[L312] ““common sense” (quote from F3)”

[L513] ““commit to a single slot” (quote from HL5)”

4) Section 1.2 - in your initial list of codes this section is called "positive IMPACT" and in the section heading later it is called "positive EXAMPLES". As "examples" I am not convinced it is really a sub-theme, but when called positive impact it makes more sense.

Author reply: Thank you. This section heading has been amended.

[L284] “1.3 Positive impact of “good” guidance”

5) Section 1.2, you mention the farrier traffic light scheme a few times - need to explain what this is

Author reply: Thank you for this. We have added a brief description of the traffic light system in order to properly introduce the concept at this stage. 

[L309] “The traffic light system utilised by farriers was a risk-based system published by the BFBA directed to provide registered farriers with guidance determining the urgency of farrier care required on a case-by-case basis.”

6) All of section 1 - you don't mention laminitis/obesity here, which is fine as I know it sets the background for the following sections. However, perhaps you could signpost this; for example before going into detail on the results you could say "firstly we will describe the way in which the guidelines affected the equestrian community generally, before considering how these impacted on equine management in relation to laminitis and obesity" (or somesuch) - just the first time I read it, I had to remind myself where I was in the paper, that's all.

Author reply: Thank you for your comment. A small paragraph ahead of the main results section has been added to clarify the order of the following results section: 

[L206] “The following sections present the themes and sub-themes in more detail. Firstly in relation to the way in which the interpretation of guidelines affected stakeholders within the equestrian community, followed by the way in which these guidelines impacted on the management of horses and ponies at greater risk of obesity and laminitis, and finally the experiences of interviewees regarding minimizing the risk of physical injury.” 

7) Section 2.2. first you mention an increase in work, but next that there was LONGER between trimming cycles, so this is a bit confusing (esp as the section is presented as if reducing visits from professionals was one thing yards implemented to reduce disease risk, but then start the section by saying they actually did more work). I think this could just be re-ordered and have a bit more explanation to clarify.

Author reply: Thank you for this recommendation. This conflict has been amended and a statement added to clarify the increased workload after the initial efforts to reduce visiting professionals on yards in the early stages. 

[L417] “The increased workload highlighted by veterinarians and farriers at the point of interview was notably due to the “backlog” (quote from F4) of work corresponding with an increased urgent need of care following initial efforts to reduce visiting professionals on yards during the early stages of lockdown.”

8) Section 2.2. - this section could be linked a bit more closely to laminitis and obesity I think? It's only mentioned in the middle

Author reply: Thank you for this observation. A short paragraph has been added to this section to highlight the important changes in laminitis and obesity management relating to changes in veterinary and hoof care. 

[L411] “The consensus amongst veterinarians was that the management of laminitis cases was not significantly impacted by practice changes. This was due to the emergency status of suspected laminitis cases, and visits to such cases were carried out as normal. However, veterinarians did comment that the less frequent interaction with horses during the early phase of lockdown accentuated cases of weight-gain and obesity.”

9) section 2.3. "restricted owner control..." - v interesting section, just not convinced the section name actually covers its contents! It partially does, but you actually focus more on yard visits/biosecurity than what the owner can or cannot do with their horse. I can see why biosecurity is relevant here, but wonder if a) you could reduce the amount you say about it so you only talk about it in reference to owner restricted control on their actual hrosecare and not more generally, and/or b) talk about yard restrictions/increasing biosecurity elsewhere

Author reply: Thank you for this recommendation. The aim of this section was intended to discuss the restrictions to horse care, however your advice has been incorporated throughout this section to emphasise the focus, and the latter portion of the section with a biosecurity focus has been removed. 

10) Line 513 authority's should be authorities

Author reply: Thank you for picking that up – it’s now been amended. 

11) section 3 - Need to explain why authorities suggested reducing activity/risk of injury, and perhaps in the intro say why this is relevant to obesity/laminitis

Author reply: Thank you for raising this. The introduction to this section has been amended to better explain the chain of events leading to horse owners reducing riding and how this is connected with laminitis and obesity. 

[L531] “Government guidelines for minimising the pressure on the NHS recommended that individuals did not engage in risk sports, such as horse riding. This guidance was reiterated by some, but not all, equestrian authorities, leaving horse owners confused over the precise rules regarding exercising horses. Some interpreted equine specific guidance as a ban on riding, whilst others saw riding as a necessary and justifiable activity for preventing equine obesity and thus minimizing the risk of laminitis”

12) 3.1 and 3.2 link more clearly to obesity/laminitis

Author reply: Thank you for this recommendation. These two sections have been developed to highlight laminitis and obesity at the core of the themes. 

[L549] “was such that it was not worth riding to potentially reduce the risk of laminitis. One welfare centre manager summarised the concept of prioritising human health over preventative horse care, saying: 

“You know, you are trying to save your own life first. Not the ponies because they are a bit more resilient that we give them credit for” – WCM1”

[L572] “As a measure to manage weight gain, some owners opted to increase the amount of “in-hand” work (exercising and training horses from the ground) as an alternative to riding during the lockdown period. This approach to exercising the horse was endorsed by the veterinarian group as a potential weight gain intervention where riding was not suitable.”

[L583] “With regards to laminitis cases, WCM1 commented that despite high levels of demand on rehoming centres to house laminitic ponies, the realities of what was achievable in the midst of the pandemic made taking these ponies on an unrealistic solution. This was in part due to the high level of close contact between handlers and veterinarians, and the intensive management, required to house a laminitic horse or pony, which may not be in the best interest of the centre or the veterinarian in terms of maintaining safe hygiene practices during the pandemic. 

“in this pandemic situation, if we were to bring in laminitics, we would have to make a serious decision on- are we going to continue with this horse or pony or not? And with so many chapping at the doors, more likely than not, we wouldn’t have to luxury of pulling everyone through it” – WCM1”

13) line 643-644 about traffic light scheme - it doesn't help owners MANAGE obesity at all, it just is supposed to help vets bring up the issue with owners (trying to help awareness)

Author reply: Thank you for clarifying this information. The wording around the traffic light scheme has been altered to reflect the use of the system for information sharing / raising awareness. 

[L665] “who implemented a colour-coded monitoring system to facilitate communication between vets and horse owners surrounding the awareness of equine obesity”.

14) line 703-705 - there must be a more recent UK example of owners looking to vets for advice? Just not sure a 1994 outbreak of a disease not v well known here is a good example.

Author reply: Thank you. This is actually a recent publication (2020), and although it refers to the emergence of the zoonotic disease that occurred some time ago, the findings describing horse owner seeking behaviour and the relationship between horse owners and veterinarians is relevant to this discussion. The sentence has been amended for clarity: 

[L720] “Research has identified that horse owners look to their veterinarians for guidance, both in regard to aspects of equine care and welfare in the competition industry [56] and as a source of biosecurity information relating to the Hendra virus (a zoonotic virus which can infect humans and horses) in Australia [47].” 

I think that's it! Well done again.

Reviewer #2: This study explores the challenges that recent lockdown events have posed to equine welfare, specifically in laminitis and obesity susceptible animals. Opinions from a range of stakeholders have been considered and this range of perspectives have contributed to the themes identified. The study highlights important issues such as the need for clarity and consistency in guidance issued and will provide a useful point of reference when considering the potential impact of further public health measures on UK equine welfare.

I enjoyed reading the paper and think it raised some interesting points. Suggestions for minor revisions are below:

Author reply: Thank you very much for your summary of our manuscript and we are very pleased that this reviewer acknowledges the important issues raised and its ability to serve as a reference point for the development of equine public health guidance. 

Introduction:

The introduction is cohesive and clearly states the aims of the project.

Author reply: Thank you

Methods:

In general the methods require some clarification.

It is stated that pre-defined questions and structured prompts are used, however in L149 you indicate that the structure and direction of interviews was determined by the participant. This seems like a contradiction.

Author reply: Thank you for this observation. It is agreed that this could be clarified, and this section has been amended. 

[L119] “Interviews were designed with pre-defined questions to guide discussions. However, in line with research methods utilising an iterative approach, interviewees were able to lead conversation in areas relating to the pandemic, laminitis, and obesity meaning that as the research evolved, certain concepts were not discussed across all interviews.” 

With the wording of questions varying, would you not expect to see differences in responses as a result? Similarly, with an iterative process how do you expect that the inclusion of new topics may have influenced results?

Author reply: Thank you for raising this valid point. It is hoped that the clarification of interview structure outlined in the methods section addresses this issue. Interviewees were able to lead discussion within the topics of laminitis, obesity, and the pandemic- as such, new topics were largely unique to the individuals experience within the subject area of discussion. 

[L119] “Interviews were designed with pre-defined questions to guide discussions. However, in line with research methods utilising an iterative approach, interviewees were able to lead conversation in areas relating to the pandemic, laminitis, and obesity meaning that as the research evolved, certain concepts were not discussed across all interviews.”

It is stated that you used a ‘targeted direct approach’ to sample participants but later state that it was a ‘pragmatic approach involving convenience’. To me this seems more like an opportunity sampling strategy within your five pre-defined groups.

Author reply: Thank you for your input regarding sampling strategy. The phrase “targeted, direct approach” has been removed and more clarity added to include the opportunistic element of sampling. 

[L140] “The sampling method employed at outset was a purposeful, pragmatic approach involving convenience, but heterogenous, sampling. Further participants were recruited as opportunity arose through connections with interviewees.”

Further detail about your sample is needed. Please clarify whether all participants with horses at livery were from different livery yards. More information on your horse owner (livery and home) samples would be beneficial – for example whether all owned leisure horses or if some were sport/competition horses. ‘Horse owner’ can cover a wide range of people in terms of socioeconomic or educational backgrounds, it would be interesting to know whether all horse owners came from similar backgrounds or whether a variety of perspectives were included. If this data wasn’t collected, the point could be added to the study limitations. Also whether the livery yards that your sample used were similar in their size/structure/pricing.

Author reply: Thank you for raising this important point. We absolutely agree that information regarding the socioeconomic and educational backgrounds is extremely important for conversations around the impact of the pandemic given the drastic difference in challenges facing different groups. Unfortunately, in the interest of sensitivity around questions of the socioeconomic status of our participants, we elected not to delve into questions in this subject and, as such, there is limited information on these characteristics of the sample. As recommended, a sentence has been added to the study limitations to reflect the fact that socioeconomic and educational factors were not included in analysis. In address to your points, a statement about the livery yard sample has been added, and Table S2 updated to include more details about the horse owner’s livery situation and horses’ level of riding / competition. 

[L154] “Livery yard stakeholders were based at different livery yards within the Aberdeenshire region. These premises were varied in terms of facilities available and type of livery packages offered, and the horse owners included leisure riders, as well as those participating in competition to county level. More detailed interviewee characteristics are available in Table S2”

[L841] “Questions around the socioeconomic and educational backgrounds of the participants were not included in the interviews, meaning that the unique impact of the pandemic between subpopulations of different backgrounds within the equine stakeholder group cannot be inferred from the results of the present study.”

In L148 could you add which type of average you calculated - mean, median etc.

Author reply: Thank you. This was the mean value and has been added for clarity. 

It is mentioned that codes were based on ‘suitable categories’, could you expand on this? Was this based on topics that came up most frequently? Similarly, the sentence spanning L181-184 is not very clear, perhaps an example would help illustrate your process

Author reply: Thank you for this observation. The concept of suitable categories has been modified to explain in full the process used during data analysis. In response to your recommendation, a table providing examples of this process has been added to the Supplementary materials (Table S3). 

[L179] “Descriptive codes were assigned to units of text which appeared meaningful. Meaningfulness was determined by the connection of the text unit to the subject of the management of obesity and laminitis, or by the demonstration of a viewpoint, opinion, example or stance of the interviewee on any subject. Categories of common or significant codes were then developed based upon either the frequency of expression, or the weight of emotion or importance of the concept to the interviewee. An example of this process can be viewed in Table S3.”

Results:

There is an error in L208 – do you mean ‘outside of’?

Author reply: Thank you. This has been amended. 

In L239 a phrase is used in quote marks. Was this a direct quote from a participant (if so it would be useful to state a participant number) or a figure of speech from the author? The same is seen in L254 and L258.

Author reply: Thank you. Details of the source of the quotes in section 1.2 have been added. 

[L257] ““glazed over” (quote from HL2)”

[L272] ““sing from the same hymn sheet” (quote from V4)”

[L276] ““you never know what you’re going into”, (quote from V4)”

[L312] ““common sense” (quote from F3)”

[L513] ““commit to a single slot” (quote from HL5)”

I agree that it is important to distinguish between perceived and actual reasons for turning away.

In L395 furlough is mentioned regarding veterinarians, could you clarify here if you mean furlough of colleagues or the general public?

Author reply: Thank you. This sentence has been modified for clarity:

[L410] “Veterinarians noted an increased workload throughout lockdown, largely due to furlough of practice staff…”

There is an error in L513 – authorities

Author reply: Thank you. This has been amended.

Discussion:

In general the manuscript could be more concise, there is repetition of content in the results and discussion that may be able to be cut down.

Author reply: Thank you – upon reviewing the discussion we have been able to condense it in order to improve clarity. 

The use of ‘mistrust’ in L701 is quite strong, perhaps concern is more appropriate.

Author reply: Thank you. This has been amended. 

You address the topic of subjectivity and generalisability in your limitations section nicely. However, I think that reference to your small sample size should be included here, as you stated that 5 participants were needed as a minimum for gaining sufficient information from a group, and in two of your five groups this figure was not reached.

Author reply: Thank you for your comments. A sentence has been included in the limitations section to cover this aspect:

[L847] “Finally, although the target of recruiting 5 participants per stakeholder group fell one participant short for two groups, the data and resultant coding generated from these groups was such that this smaller sample size was considered to have captured sufficient information to assume thematic saturation.”

Reviewer #3: An interesting study and good to see researchers determining the impact of the pandemic on equine welfare. Your work would add to the growing body of work in this area but currently it feels like a more general review of the impact of the pandemic on equine welfare is being shoehorned to focus on laminitis and obesity, and for me this detracts from the quality of your work and would suggest rebranding with a more general lens of the impact on ‘equine health and welfare’. I would also urge the authors to consider if a more local/ regional perspective would better represent their sample as this would provide a detailed community level response (as limited – 2 – not in a local radius).

Author reply: Thank you for your overview and appreciation of this important topic. The primary research interests of our group involve laminitis and obesity, and the current study was designed to capture the impact of the pandemic on those obesity and laminitis-prone animals by focusing on owners of native breed horses and ponies at greater risk of these conditions. However, we acknowledge that at times the manuscript may have seemed more focused on health and welfare more generally. During the subsequent revision of the manuscript, laminitis and obesity has been placed at the centre of our results and discussion and feel that the manuscript has been greatly improved as a result and we hope that you agree. 

Comments

Title

While obesity and laminitis were some key concerns arising from your interviews they were not the key focus of the interview approach taken and I feel your work is a more general review of the impact of the pandemic on equine welfare and your title should reflect this

Author reply: Thank you for your comment. As we said above, we feel that the revision of the manuscript has focused on the impact of the pandemic on the issues pertaining to caring for animals at greater risk of laminitis and obesity and therefore we feel that the title better reflects the revised content and hope you agree. 

Abstract

It would be beneficial to include a more general summary of the themes that arose from your analysis then to highlight laminitis etc as this would be more representative of the work undertaken

Author reply: Thank you. This has been incorporated into the abstract. 

 [L28] “Restrictions were likely to have resulted in collateral consequences for the health and welfare of horses and ponies, especially those at risk of obesity and laminitis and this issue warranted more detailed exploration.”

Line 30: there are a number of published studies which have considered the impact of Covid on horses which include welfare aspects, therefore suggest amending this statement to something like warrants exploration

Author reply: Thank you. This has been amended.

Line 34: suggest amending to laminitis were:

Author reply: Thank you. This has been amended.

Line 41: would be good to identify a context for the guidelines referred to e.g. management broadly within the pandemic or specifically for at risk horses / ponies

Author reply: Thank you. This sentence has been amended:

[L42] : “These findings support the development of guidelines specific to the care of horses and ponies at risk of obesity and laminitis through collaborative input from veterinary and welfare experts, to reduce the negative impacts of future lockdown events upon equine welfare in the UK.”

Introduction

Rationale for study provided complemented by background to the pandemic and its impact in horse owners.

Line 60: please remove comma after associated

Author reply: Thank you. This has been done. 

Line 62: suggest inserting: horse owners after some

Author reply: Thank you. This has been done. 

Materials and methods

You could present the methodological framework that underpins your analytical approach and interpretation more explicitly in the opening paragraph.

Author reply: Thank you. The opening paragraph has been amended to better present the methodology.

[L116] “The study was designed and conducted within the methodological framework of hermeneutic phenomenology. The grounded theory aspect of the methodological approach was consistent in the inductive, comparative approach to analysis, coding of text and verification of themes which arose from the data [17].”

Please define your inclusion / exclusion criteria for potential participants; it would be good to align these to your 5 key stakeholder groups

Author reply: Thank you for this recommendation. A table (Table S1) has been included to provide a summary of inclusion and exclusion criteria 

Your sample is largely local (Aberdeenshire) – given the differences between UK regions with respect to lockdown and quarantine regulations, I would advise the authors to consider removing the Somerset and Blackpool participants to give your work a local or Scottish focus. If you elect not to then the potential limitations of the sampling strategy should be considered in your discussion.

Author reply: Thank you for your suggestion. The decision has been made to include the Somerset and Blackpool participants. We entirely agree with your point regarding having a local focus, but we feel that due to the relatively low number of registered Equine Welfare Centres in the region, our geographical extension to increase our sample number within this group was justified, as they were all from the same welfare charity. As recommended, we have added a sentence to summarise the limitations of this sampling strategy in the discussion. 

[L835] “Applicability of the findings may also be limited by the inclusion of two respondents (WCM3 and WCM4) who were based outside of the Aberdeenshire region. It is possible that the views and experiences of these two participants may not compare to those of welfare managers based in the Aberdeenshire locale…”

Please include details of who conducted the interviews and outline their experience with this process. Did the interviewer take notes during the interview? If yes, please discuss the limitations of this approach.

Author reply: Thank you for your advice. Details of the interviewer and their experience has been added to the methods section, and the limitations of note taking during interviews has also been added to in the discussion. 

[L122] “The primary researcher, a PhD student, received training in qualitative research methods prior to and throughout the study before conducting a pilot interview which was recorded and assessed to determine the appropriateness of questions, interview style and overall approach.”

[L851] “As interviews were conducted by telephone, capturing the nuances of communication during face-to-face interview techniques was not possible. Despite this limitation, the use of telephone interviews allowed extensive notes to be taken during conversations without disrupting the fluency of the interview.” 

Line 148 – 149: please edit sentence to enhance flow as it is a bit disjointed

Author reply: Thank you. This has been amended:

[L159] “In total, 13 hours of interviews were conducted, with a mean interview length of 32 minutes (min = 17 mins, max = 45 mins).”

Line 167: suggest phrasing as Supplementary file 1: Table S1

Author reply: Thank you. This has been amended in line with the PLOS ONE author guidelines. 

Line 178 it would be beneficial to reiterate the methodological framework applied across coding within this paragraph and to add citations to support the approach taken (refer to Braun and Clarke’s work)

Author reply: Thank you for this recommendation. The following passage has been incorporated into the methodology section to improve the clarity of the coding process with supporting references. 

[L178] “Data were organised and interpreted using an iterative coding process directed by a hermeneutic approach to analysis [20]. Descriptive codes were assigned to units of text which appeared meaningful. Meaningfulness was determined by the connection of the text unit to the subject of the management of obesity and laminitis, or by the demonstration of a viewpoint, opinion, example or stance of the interviewee on any subject. Categories of common or significant codes were then developed based upon either the frequency of expression, or the weight of emotion or importance of the concept to the interviewee [21]. An example of this process can be viewed in Table S3.”

Please can you clarify how many of the research team analysed the results and if triangulation occurred at this level or was a review of the summary table only

Author reply: Thank you. The details of involvement in analysis and triangulation have been added. 

[L191] Analysis was carried out by the primary researcher, and tables of units, codes, sub themes and themes was recorded and shared amongst the research team to undergo discussion of appropriateness at multiple stages during the data analysis phase of the study. Such reviews were conducted to improve the validity of the theory being drawn from the interview transcripts, whilst maximising efficiency amongst the group. 

Did you engage in any stakeholder verification of the results?

Author reply: Thank you. Stakeholder verification of the results was not performed in the present study, however, we agree that this would serve to improve the soundness of the results. 

Results

Detailed discussion of themes and sub themes presented; does overall feel a little bit negative and I wonder if there are some more positive aspects which you could highlight to balance the selected quotes more.

Author reply: Thank you for this observation. We agree that the results could have a negative tone in places. We hope that this has been improved through the addition of quotes with more optimistic tone to provide a better evaluation of the results. 

[L236] “Interestingly, the EWC group perceived the multiple sources of information as less of an issue, but as having the potential for positive impacts on information exchange. One welfare centre manager outlined the connections between the organisations and highlighted the veterinary associations as the welfare centre’s ultimate point of reference for information, saying: 

“you have NEWC, the National Equine Welfare Council, of which we are part of. That is basically equine charities which have all come together…. although we all work individually, we all know what each other are doing at one time. Obviously, the British Equestrian Federation, and BEVA, the veterinary advisory groups as well…our head office keeps abreast of exactly what they are going to issue, and we have to follow their guidelines.” – WCM1”

[L278] “There were also points in conversation where veterinarians indicated that they felt prepared for the pandemic in a sense, due their training in infectious disease and epidemiology. 

“I have no doubt that the RCVS (Royal College of Veterinary Surgeons) and BVA (British Veterinary Association) have been setting guidance, and also my employer who is a corporate employer, they would give out instructions every day… I was just following the simple biosecurity measures that we are always aware of,”- V2”

Line 191: I think identifying your 4 key themes here would be beneficial

Author reply: Thank you. A supplementary figure (Figure S1) has now been generated to summarise the codes, sub-themes and higher-order themes and how they link together. 

Line 195: I would consider your presentation of the higher/ lower order themes as a list – I think a table or figure would enhance the impact of your work (and include theme 4); it may be nice to include some contextualisation to add flavour to the theme / sub themes presented as well

Author reply: Thank you – see above reply. 

Line 208: please edit sentence suggest is out of the scope of the present article

Author reply: Thank you. The sentence has been amended:

[L203] “The authors considered that detailed exploration of the theme of mental health during the pandemic was out-with the scope of the present article and the data will be presented in a separate manuscript.”

Discussion

Interesting points debated but the discussion is a little lengthy and in parts a little repetitious to results; would suggest an edit to make more concise would be beneficial to increase impact

It would be of interest to evaluate if the approach to the pandemic perceived by equine stakeholders is unique or if similar patterns occurred across other groups e.g. small animal vets, dog owners, animal shelters, homeless shelters etc reflecting a more global pandemic phenomenon, within your discussion.

Author reply: Thank you. The discussion has been condensed and edited to improve clarity and to better place the study within wider COVID-19 related literature. 

[L676] “as well as concerns from a welfare perspective around the increased demand on rehoming as an inevitable result of the pandemic”

[L700] “Given the relationship shown between turning horses away and the furlough of key equestrian staff, it may also be beneficial to consider the benefit that emergency legislation and funding to support key workers involved in the care of equids at risk of laminitis would have during emergency scenarios. Funding during the pandemic was identified as the most frequently cited concern of third sector organisations in a survey conducted by the Scottish Government [49]. Financial aid to protect those most in need during the COVID-19 pandemic was actioned in Scotland, with support services for key vulnerable groups receiving part of the Scottish Government’s £350 million Communities Funding reserve [50]. In the equestrian industry, The Petplan Charitable Trust (PPCT), World Horse Welfare and National Equine Welfare Council collaborated to offer grants of £5000 for organisations involved in the rescue and rehoming of equines [51], and support for small equine businesses was available from the Government [52]. Considering the cost of upkeep of horses, estimated at £3,105 per horse per year in the UK [53], additional costs related with the diagnosis, treatment and prevention of laminitis and obesity could make caring for a laminitic horse an unfeasible expense for some. Thus, the allocation of emergency funding for the costs associated with laminitis care may help to support horses, equestrian businesses and workers in the equestrian industry in the future.”

[L715] “As was seen in human medicine and small animal veterinary practice [54]”

Line 569: please replace this with these

Author reply: Amended

Line 647/8: there are examples of positive practices across different industries including equestrianism – may not be the headlines in research but they are there – also suggest reviewing some of the grey literature and industry sources in respect of this

Author reply: Thank you for highlighting this. The phrase has been omitted. In addition, we hope that the response to your previous comment, which incorporates various sources of information in grey literature regarding funding support, highlights our appreciation of your comment that this field is, indeed, rich with positive practice examples. 

Lines 655 / 663: please remove highlighted text

Author reply: Done

Conclusions

For me your conclusions should relate to the key findings of your work i.e. the 4 themes and subthemes as the headline and then you can link to laminitis and obesity – suggest reconsidering this section in line with broader feedback to shift the focus of the paper to broader equine health / welfare.

Author reply: Thank you. The conclusion has been altered to address this.

---

## [Decision Letter · Decision Letter 1]

14 May 2021

COVID-19 impacts equine welfare: policy implications for laminitis and obesity

PONE-D-20-38153R1

Dear Dr. Morrison,

We’re pleased to inform you that your manuscript has been judged scientifically suitable for publication and will be formally accepted for publication once it meets all outstanding technical requirements.

Kind regards,

Ludek Bartos

Academic Editor

PLOS ONE

Additional Editor Comments (optional):

Reviewers' comments:

Reviewer's Responses to Questions

**Comments to the Author**

1. If the authors have adequately addressed your comments raised in a previous round of review and you feel that this manuscript is now acceptable for publication, you may indicate that here to bypass the “Comments to the Author” section, enter your conflict of interest statement in the “Confidential to Editor” section, and submit your "Accept" recommendation.

Reviewer #1: All comments have been addressed

Reviewer #2: All comments have been addressed

Reviewer #3: (No Response)

2. Is the manuscript technically sound, and do the data support the conclusions?

Reviewer #1: Yes

Reviewer #2: Yes

Reviewer #3: Yes

3. Has the statistical analysis been performed appropriately and rigorously? 

Reviewer #1: N/A

Reviewer #2: Yes

Reviewer #3: Yes

4. Have the authors made all data underlying the findings in their manuscript fully available?

Reviewer #1: Yes

Reviewer #2: Yes

Reviewer #3: Yes

5. Is the manuscript presented in an intelligible fashion and written in standard English?

Reviewer #1: Yes

Reviewer #2: Yes

Reviewer #3: Yes

6. Review Comments to the Author

Reviewer #1: I am satisfied that the manuscript has been updated based on the comments of all three reviewers, and my recommendation would be that it is now accepted. Well done to the team.

Reviewer #2: (No Response)

Reviewer #3: Thank you for submitting your revised manuscript, the revisions made have increased the impact of your work. This study adds another dimension to support broader interpretation of the impact of the current coronavirus pandemic on equine welfare and as such will be of interest to readers and will help inform ongoing industry responses to the situation. Pleased to recommend accepting for publication - couple minor points to consider below.

625 - closing bracket missing

670 - later than egs provided but there are now equine yard specific Covid guidelines supported by BHS, NEWC, BEVA and others which could be worth identifying in here

7. PLOS authors have the option to publish the peer review history of their article (what does this mean?). If published, this will include your full peer review and any attached files.

Reviewer #1: **Yes: **Tamzin furtado

Reviewer #2: No

Reviewer #3: No

---

## [Editor Report · Acceptance letter]

19 May 2021

PONE-D-20-38153R1 

COVID-19 impacts equine welfare: policy implications for laminitis and obesity 

Dear Dr. Morrison:

I'm pleased to inform you that your manuscript has been deemed suitable for publication in PLOS ONE. Congratulations! Your manuscript is now with our production department. 

Kind regards, 

on behalf of

Dr. Ludek Bartos 

Academic Editor

PLOS ONE